# Dive3D: Diverse Distillation-based Text-to-3D Generation via Score Implicit Matching

**Weimin Bai**                                                                    *weiminbai@stu.pku.edu.cn*
*Academy for Advanced Interdisciplinary Studies, Peking University*
*National Biomedical Imaging Center, Peking University*

**Yubo Li**                                                                       *ybli25@stu.pku.edu.cn*
*School of Software and Microelectronics, Peking University*
*National Biomedical Imaging Center, Peking University*

**Wenzheng Chen**                                                                *wenzhengchen@pku.edu.cn*
*Wangxuan Institute of Computer Technology, Peking University*

**Weijian Luo**                                                                   *pkulwj1994@icloud.com*
*hi-lab, Xiaohongshu Inc*

**He Sun**[*]                                                                     *hesun@pku.edu.cn*
*Future of Technology, Peking University.*
*National Biomedical Imaging Center, Peking University*

**Reviewed on OpenReview:** *https://openreview.net/forum?id=OUYMueHLMf*

## Abstract

Distilling pre-trained 2D diffusion models into 3D assets has driven remarkable advances in text-to-3D synthesis. However, existing methods typically rely on Score Distillation Sampling (SDS) loss, which involves asymmetric KL divergence—a formulation that inherently favors mode-seeking behavior and limits generation diversity. In this paper, we introduce Dive3D, a novel text-to-3D generation framework that replaces KL-based objectives with Score Implicit Matching (SIM) loss, a score-based objective that effectively mitigates mode collapse. Furthermore, Dive3D integrates both diffusion distillation and reward-guided optimization under a unified divergence perspective. Such reformulation, together with SIM loss, yields significantly more diverse 3D outputs while improving text alignment, human preference, and overall visual fidelity. We validate Dive3D across various 2D-to-3D prompts and find that it consistently outperforms prior methods in qualitative assessments, including diversity, photorealism, and aesthetic appeal. We further evaluate its performance on the GPTEval3D benchmark, comparing against nine state-of-the-art baselines. Dive3D also achieves strong results on quantitative metrics, including text–asset alignment, 3D plausibility, text–geometry consistency, texture quality, and geometric detail.

## 1 Introduction

Text-to-3D generation—the task of creating 3D contents from natural language descriptions—has attracted enormous interest Poole et al. (2022); Shi et al. (2023); Lin et al. (2023), due to its broad applications in vision and graphics. Recent advances, such as 3D representations(Mildenhall et al., 2021; Kerbl et al., 2023), large-scale pre-trained vision-language models(Radford et al., 2021), advanced text-to-image diffusion and flow models(Rombach et al., 2022), and differentiable rendering techniques, have further accelerated progress in this field. In particular, powerful text-to-image diffusion models such as Stable Diffusion series(Rombach

---

[*]Corresponding author.

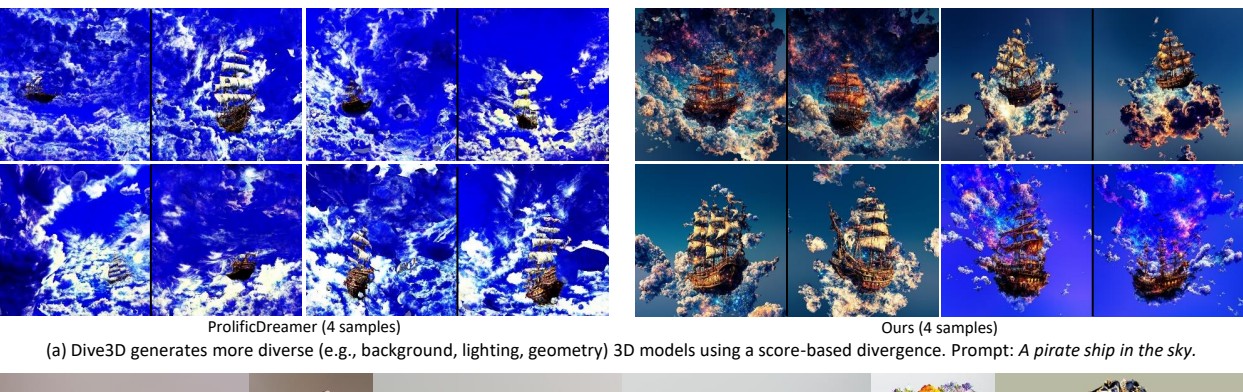

ProlificDreamer (4 samples)  Ours (4 samples)

(a) Dive3D generates more diverse (e.g., background, lighting, geometry) 3D models using a score-based divergence. Prompt: *A pirate ship in the sky.*

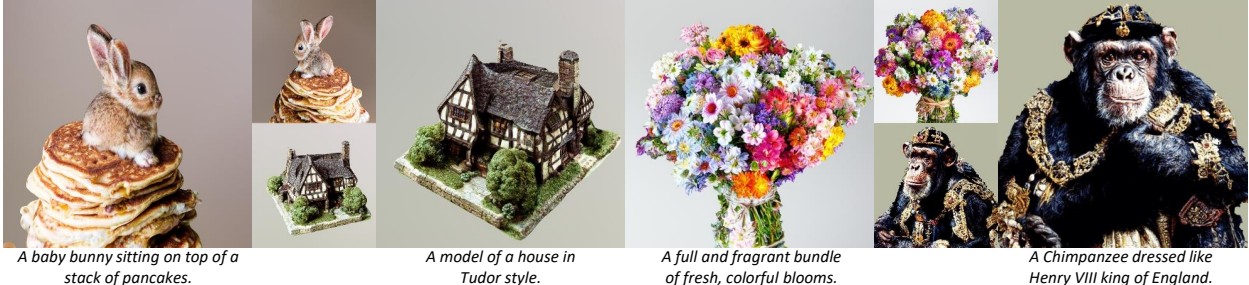

*A baby bunny sitting on top of a stack of pancakes.*  *A model of a house in Tudor style.*  *A full and fragrant bundle of fresh, colorful blooms.*  *A Chimpanzee dressed like Henry VIII king of England.*

(b) Dive3D unifies SDS and reword guidance, producing higher-quality and human-preferred 3D models.

Figure 1: We propose **Dive3D**, a novel text-to-3D generation framework that enables both **diverse** and **high-fidelity** 3D content creation. **Top**: Compared to ProlificDreamer, which exhibits mode collapse under the same prompt, Dive3D generates a diverse set of realistic and semantically aligned 3D outputs. **Bottom**: Dive3D unifies SDS and human-preference rewards within a divergence-based framework, effectively combining diffusion priors and reward signals to boost both visual quality and prompt alignment. All results are represented as textured mesh assets.

et al., 2022; Ramesh et al., 2022; Saharia et al., 2022), lay a strong foundation for text-driven 3D synthesis: by leveraging pre-trained 2D diffusion priors and multi-view rendering, one can optimize a 3D asset so that its renderings align with a given text prompt. This capability opens new avenues for 3D content creation, enabling even non-experts to "describe and create" novel 3D assets in freestyles.

Several paradigms have emerged to tackle text-to-3D generation. Diffusion distillation-based methods—exemplified by Score Distillation Sampling (SDS) in DreamFusion Poole et al. (2022)—optimize 3D representations by aligning multi-view renderings with pre-trained text-to-image diffusion priors Rombach et al. (2022). Reward-guided approaches Mohammad Khalid et al. (2022); Jain et al. (2022) further refine these approaches by directly incorporating human-preference or CLIP-based rewards, boosting both semantic alignment and perceived quality. Despite their impressive fidelity and text alignment, both diffusion-distillation and reward-guided methods suffer from a critical limitation: *limited generative diversity*. Even when prompted with intentionally vague or open-ended descriptions, current models tend to converge on a narrow set of similar outputs.

We analyze this limitation and trace its root to the utilization of Kullback–Leibler (KL) divergence. Specifically, the objectives optimized by both SDS and reward-based methods can be reformulated as minimizing an asymmetric KL divergence, which shares a fundamental limitation: KL divergence inherently encourages mode-seeking behavior by penalizing samples that deviate from high-density regions of the target distribution. As a result, the generative model tends to collapse to a few dominant modes, severely suppressing output diversity.

In this paper, we present Dive3D, a novel framework that replaces KL-based objectives with Score Implicit Matching (SIM)—a score-based divergence loss that directly matches the gradient fields of the probability density of generated contents and the diffusion prior. This formulation avoids the mode-seeking tendencies of KL and encourages exploration of multiple high-probability regions, thereby promoting diversity without

sacrificing fidelity or alignment. Furthermore, Dive3D unifies both diffusion distillation and reward-guided optimization under a divergence-based perspective. Combined with SIM loss, this formulation enables a principled integration of diffusion priors, human preferences, and diversity-promoting objectives within a single framework. As a result, Dive3D generates 3D assets that are not only realistic and well-aligned with text prompts, but also significantly more diverse.

Through extensive experiments on standard text-to-3D benchmarks, we demonstrate that Dive3D achieves state-of-the-art performance, substantially outperforming existing SDS- and reward-based approaches across visual fidelity, prompt adherence, and generative diversity.

## 2 Related Works

**Diffusion Distillation Based Methods**  Diffusion distillation methods (Luo, 2023) leverage pre-trained text-to-image diffusion models (Rombach et al., 2022; Saharia et al., 2022; Balaji et al., 2022) to guide the optimization of 3D representations by aligning rendered views with diffusion priors. This line of work was pioneered by Score Distillation Sampling (SDS) in DreamFusion (Poole et al., 2022), which initiated an era of 3D synthesis by transferring the knowledge embedded in 2D diffusion priors. However, these diffusion-driven optimization techniques typically rely on minimizing KL divergences (Poole et al., 2022; Wang et al., 2024; Luo et al., 2024a), often resulting in mode-seeking behavior where generated 3D objects collapse into a single plausible solution with limited diversity. Moreover, the straightforward use of 2D diffusion priors can introduce visual artifacts such as over-saturated colors, overly smooth geometry, and even Janus artifacts (Jain et al., 2022; Chen et al., 2023; Wang et al., 2024). To address these challenges, recent studies have explored various improvements, including timestep annealing (Huang et al., 2024a; Wang et al., 2024; Zhu et al., 2024), coarse-to-fine training (Lin et al., 2023; Wang et al., 2024; Chen et al., 2023), component analysis (Katzir et al., 2024), and formulation refinements (Zhu et al., 2024; Wang et al., 2024; Liang et al., 2023; Tang et al., 2023a; Wang et al., 2023b; Yu et al., 2024; Armandpour et al., 2023; Wu et al., 2024b; Yan et al., 2024). Additional efforts have focused on geometry-texture disentanglement (Chen et al., 2023; Ma et al., 2023a; Wang et al., 2024) and mitigating the multi-face (Janus) problem by replacing text-to-image diffusion with novel view synthesis or multi-view diffusion (Liu et al., 2023; Long et al., 2023; Liu et al., 2024; Weng et al., 2023; Ye et al., 2023; Wang & Shi, 2023; Shi et al., 2023). Notably, diffusion distillation has also seen rapid progress in other domains, such as one-step diffusion models (Luo et al., 2023a; 2024b; Yin et al., 2024; Zhou et al., 2024b; Geng et al., 2024; Huang et al., 2024b) and various related approaches (Nie et al., 2022; Zhang et al., 2023a;b; Luo et al., 2023b).

**Reward Optimization based Methods.**  Another category of approaches optimizes 3D outputs directly using reward models, such as visual-language alignment losses or human-preference reward models instead of (or in addition to) a diffusion prior. Early methods like CLIP-Mesh (Mohammad Khalid et al., 2022) and DreamFields (Jain et al., 2022) directly maximize the CLIP score (Radford et al., 2021) between rendered images and the text prompt, enabling zero-shot text-to-3D without 3D datasets. While conceptually simple, these CLIP-guided approaches often yielded suboptimal geometry or texture (e.g. unrealistic shapes) and required expensive optimization. More recently, DreamReward (Ye et al., 2024) uses a learned internal 3D preference-reward model (Reward3D) trained on internally collected human feedback data to guide generation. DreamReward improves alignment of generated shapes with user intent, achieving better text relevance as judged by the reward function. Reward-based methods explicitly push for semantic or aesthetic alignment, but relying solely on them can compromise visual fidelity if the reward is not perfectly aligned with 3D realism (e.g. CLIP might encourage implausible textures). They may also require costly human data collection to train the internal 3D reward model.

**Feed-forward Methods.**  Feed-forward methods train neural networks to directly generate 3D content from text using large synthetic 3D datasets or cross-modal supervision. For example, CLIP-Forge (Sanghi et al., 2022) and CLIP-Sculptor (Sanghi et al., 2023) leverage CLIP embeddings for zero-shot text-to-shape generation. More recently, advances in large reconstruction models (LRMs)(Hong et al., 2023) have enabled rapid 3D model prediction from single or sparse-view images, inspiring these developments of methods like Instant3D(Li et al., 2023) and Turbo3D (Hu et al., 2024) that first generate multi-view images from text and

then use a feed-forward 3D reconstructor (trained on synthetic data) to instantly produce representations such as NeRF or 3D Gaussian Splatting. However, the quality of these approaches depends heavily on the underlying text-to-multi-view generator, often recasting the challenge as one of diffusion distillation or reward-based optimization.

## 3 Preliminary

In this section, we review the key concepts and mathematical formulations underlying our work. We first describe text-to-image diffusion models, then explain how these models are adapted for text-to-3D generation via diffusion distillation, and finally review reward-guided text-to-3D methods.

### 3.1 Text-to-Image Diffusion Models

Diffusion models Sohl-Dickstein et al. (2015); Ho et al. (2020); Song et al. (2020b) are a class of generative models that iteratively transform noise into data using a stochastic process. Let $\boldsymbol{x}_0 \sim q_{\text{data}}(\boldsymbol{x})$ denote a data sample. The forward diffusion process corrupts $\boldsymbol{x}_0$ by gradually adding noise described by the stochastic differential equation (SDE):

$$d\boldsymbol{x}_t = \boldsymbol{F}(\boldsymbol{x}_t, t)\, dt + G(t)\, d\boldsymbol{w}_t, \quad t \in [0, T], \tag{1}$$

where $\boldsymbol{F}(\boldsymbol{x}_t, t)$ is a drift function, $G(t)$ is a scalar-valued diffusion coefficient, and $\boldsymbol{w}_t$ denotes a standard Wiener process. To generate samples, the reverse diffusion process is used to progressively denoise an initial noise sample (Song et al., 2020b;a; Zhang & Chen, 2022; Liu et al., 2022; Lu et al., 2022; Xue et al., 2023).

The marginal core function $\nabla_{\boldsymbol{x}_t} \log p_t(\boldsymbol{x}_t)$ is typically approximated by a continuous-indexed neural network $s_\phi(\boldsymbol{x}_t, t)$. This score network is trained using the weighted denoising score matching objective:

$$\mathcal{L}(\phi) = \mathbb{E}_{t, \boldsymbol{x}_0, \epsilon} \left[ \lambda(t) \left\| s_\phi\left( \alpha_t \boldsymbol{x}_0 + \sigma_t \epsilon, t \right) + \frac{\epsilon}{\sigma_t} \right\|_2^2 \right], \tag{2}$$

where $\epsilon \sim \mathcal{N}(0, \mathbf{I})$, and the functions $\alpha_t$ and $\sigma_t$ are determined by the noise schedule.

By conditioning on text inputs, these diffusion models can be extended to text-to-image synthesis. In this setting, a conditional score network $s_\phi(\boldsymbol{x}_t, y, t) \approx \nabla_{\boldsymbol{x}_t} \log p_t(\boldsymbol{x}_t | y)$ is used, where $y$ is the text prompt describing the image content. Popular models such as Stable Diffusion (Rombach et al., 2022) and MVDiffusion (Shi et al., 2023) have demonstrated that this approach yields high-quality, semantically aligned images.

### 3.2 Text-to-3D Generation by Diffusion Distillation

A prevalent paradigm for text-to-3D synthesis leverages pretrained text-to-image diffusion models to guide the optimization of a 3D representation. Let $g(\theta, c)$ be a differentiable renderer that maps the 3D parameters $\theta$ to a 2D image under camera pose $c$, $q_\theta(\boldsymbol{x}_t | c)$ be the distribution of rendered images at diffusion time $t$, and $p(\boldsymbol{x}_t | y^c)$ be the target conditional distribution given a view-dependent text prompt $y^c$ defined by a pretrained diffusion model. The loss that aligns each rendered view of the 3D model with the conditional diffusion prior can be formulated as:

$$\mathcal{L}_{\text{CDP}}(\theta) = \mathbb{E}_{t, c} \left[ \omega(t)\, D_{\text{KL}}\left( q_\theta(\boldsymbol{x}_t | c)\, \big\|\, p(\boldsymbol{x}_t | y^c) \right) \right], \tag{3}$$

where $\omega(t)$ is a weighting function. In practice, the gradient of loss (3) writes (please refer to Luo et al. (2024a) and Wang et al. (2023a) for a comprehensive derivation):

$$\nabla_\theta \mathcal{L}_{\text{CDP}}(\theta) \approx \mathbb{E}_{t, \epsilon, c} \left[ \omega(t)\, (\epsilon_\phi(\boldsymbol{x}_t, y^c, t) - \epsilon)\, \frac{\partial g(\theta, c)}{\partial \theta} \right], \tag{4}$$

where $\epsilon_\phi(\boldsymbol{x}_t, y^c, t) = -\sigma_t s_\phi(\boldsymbol{x}_t, y^c, t)$ is the noise prediction of the diffusion model.

The Score Distillation Sampling (SDS) loss, introduced in DreamFusion Poole et al. (2022), improves generation quality by employing classifier-free guidance (CFG)(Ho & Salimans, 2022; Ahn et al., 2024; Karras

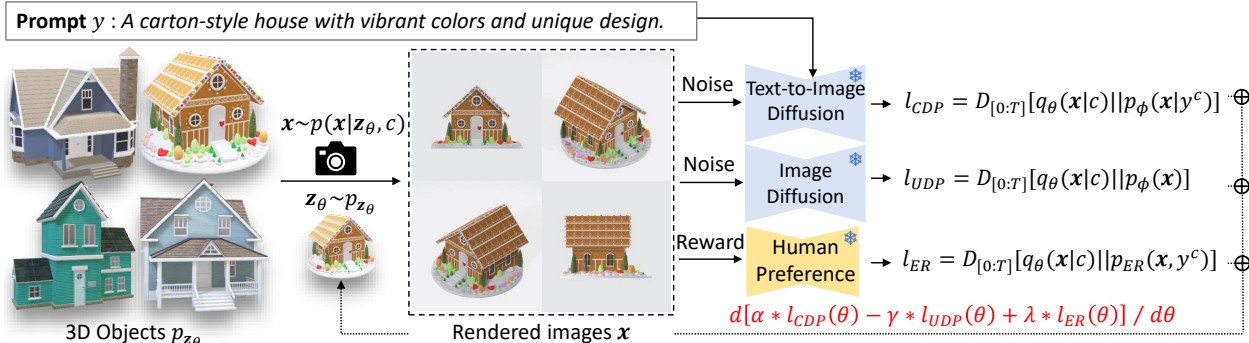

Figure 2: **Overview of Dive3D.** Dive3D reformulates both SDS loss and human-preference rewards within a unified divergence-based framework, revealing that both rely on KL divergence, $D_{\mathrm{KL}}(\cdot, \cdot)$—an asymmetric objective that inherently favors mode-seeking and leads to model collapse and limited diversity. Dive3D replaces this with a **Score Implicit Matching**-based divergence, $D_{[0,T]}(\cdot, \cdot)$, which aligns score fields rather than probability densities, effectively mitigating mode collapse and enabling more diverse and faithful 3D generation.

et al., 2024; Li et al., 2024), which replaces the original conditional score in Eq.4 with a weighted difference between the conditional and unconditional score estimates,

$$
\begin{aligned}
\hat{\epsilon}_\phi(\boldsymbol{x}_t, y^c, t) &= (1 + \gamma)\epsilon_\phi(\boldsymbol{x}_t, y^c, t) - \gamma\,\epsilon_\phi(\boldsymbol{x}_t, t) \\
&= \epsilon_\phi(\boldsymbol{x}_t, y^c, t) + \gamma\Big(\epsilon_\phi(\boldsymbol{x}_t, y^c, t) - \epsilon_\phi(\boldsymbol{x}_t, t)\Big) \\
&= -\sigma_t\Big[s_\phi(\boldsymbol{x}_t, y^c, t) + \gamma\Big(s_\phi(\boldsymbol{x}_t, y^c, t) - s_\phi(\boldsymbol{x}_t, t)\Big)\Big],
\end{aligned}
\tag{5}
$$

$$
\nabla_\theta \mathcal{L}_{\mathrm{SDS}}(\theta) \approx \mathbb{E}_{t,\epsilon,c}\left[\omega(t)\left(\hat{\epsilon}_\phi(\boldsymbol{x}_t, y^c, t) - \epsilon\right)\frac{\partial g(\theta, c)}{\partial \theta}\right].
\tag{6}
$$

This adjustment is equivalent to incorporating an additional regularization term into the Score Distillation Sampling (SDS) loss - the so-called CFG-reward as introduced by (Luo et al., 2024c) ($\mathcal{L}_{\mathrm{SDS}} = \mathcal{L}_{\mathrm{CDP}} + \gamma\mathcal{L}_{\mathrm{CFR}}$), effectively acting as an implicit likelihood term that better aligns the generated image with the text prompt and enforces pose constraints. Increasing the weighting factor $\gamma$ strengthens this alignment, thereby improving the semantic calibration of the 3D renderings.

### 3.3 Text-to-3D Generation via Reward Guidance

An alternative approach leverages reward signals to steer the generation of 3D content. Pioneering works such as DreamFields Jain et al. (2022), CLIP-Mesh Mohammad Khalid et al. (2022), and X-Mesh Ma et al. (2023b) leverage CLIP scores Radford et al. (2021) to align 3D representations with text prompts. In these methods, a reward function is defined as:

$$
r(y, x, c) = f\big(g(\theta, c)\big)^\top h(y^c),
\tag{7}
$$

where $f(\cdot)$ and $h(\cdot)$ are embedding functions for images and text, respectively, and $g(\theta, c)$ is the rendered image. Maximizing this reward encourages the 3D model to generate outputs that are semantically aligned with the text.

Recent methods, such as DreamReward Ye et al. (2024), combine the SDS loss with reward-based signals to further enhance semantic alignment and human-preference consistency. For example, DreamReward modifies the SDS loss as:

$$
\mathcal{L}_{\mathrm{Reward}}(\theta) = \mathcal{L}_{\mathrm{SDS}}(\theta) - \lambda\,\mathbb{E}_{t,c,\boldsymbol{x}_t}\Big[\omega(t)\,r\big(y^c, \hat{x}_0(\boldsymbol{x}_t)\big)\Big],
\tag{8}
$$

where $\boldsymbol{x}_t \sim q_\theta(\boldsymbol{x}_t|c)$, $\hat{x}_0 = \frac{1}{\alpha_t}\left[\boldsymbol{x}_t - \sigma_t\epsilon_\phi(\boldsymbol{x}_t, y, t)\right]$ is an estimate of the denoised image, and $\lambda$ balances the influence of the reward. Similar to Eq. 5, the reward function acts as an additional regularization term in SDS-based 3D generation.

## 4 Method

In this section, we introduce *Dive3D*, a principled framework that boosts both diversity and fidelity in text-to-3D synthesis by replacing KL-divergence guidance with score-based divergence optimization (see Fig.2). In Sec.4.1, we demonstrate that existing SDS and reward losses are both linear combinations of KL divergences—and thus prone to mode collapse and mode-seeking. Then, in Sec. 4.2, we present our score-based divergence formulation, which overcomes these limitations and delivers significantly more varied and higher-quality 3D outputs.

### 4.1 SDS and Reward Guidance are Both KL Divergences

**The SDS Loss.** The classifier-free guidance in the SDS loss (Eqs. 5–6) can be rewritten as

$$s_\phi(\boldsymbol{x}_t, y, t) - s_\phi(\boldsymbol{x}_t, t) \approx \nabla_{\boldsymbol{x}_t} \log p(\boldsymbol{x}_t | y) - \nabla_{\boldsymbol{x}_t} \log p(\boldsymbol{x}_t). \tag{9}$$

Substituting Eq. 9 into Eq. 6 and integrating, the SDS loss can be expressed as the difference between two KL divergence terms:

$$
\begin{aligned}
\mathcal{L}_{\text{SDS}}(\theta) &= (1+\gamma)\,\mathcal{L}_{\text{CDP}}(\theta) - \gamma\,\mathcal{L}_{\text{UDP}}(\theta) \\
&= (1+\gamma)\,\mathbb{E}_{t,c}\left[\omega(t)\,D_{\text{KL}}\Big(q_\theta(\boldsymbol{x}_t|c)\,\big\|\,p(\boldsymbol{x}_t|y^c)\Big)\right] \\
&\quad - \gamma\,\mathbb{E}_{t,c}\left[\omega(t)\,D_{\text{KL}}\Big(q_\theta(\boldsymbol{x}_t|c)\,\big\|\,p(\boldsymbol{x}_t)\Big)\right].
\end{aligned}
\tag{10}
$$

This formulation makes explicit that the SDS loss balances two KL divergences—one that promotes prompt fidelity ($\mathcal{L}_{\text{CDP}}$) and one that modulates diversity via the unconditional prior ($\mathcal{L}_{\text{UDP}}$). Increasing $\gamma$ strengthens text–image alignment but narrows diversity by shrinking the effective entropy.

**The Explicit Reward Loss.** Assuming the reward defines an exponential distribution,

$$p_{\text{ER}}(y^c, x_t) \propto \exp\Big(r\big(y^c, \hat{x}_0(\boldsymbol{x}_t)\big)\Big), \tag{11}$$

the explicit reward loss in Eq. 8 can likewise be interpreted as a KL divergence.

$$
\begin{aligned}
\mathcal{L}_{\text{ER}}(\theta) &= \mathbb{E}_{t,c}\left[\omega(t)\,D_{\text{KL}}\Big(q_\theta(\boldsymbol{x}_t|c)\,\big\|\,p_{\text{ER}}(y^c, \boldsymbol{x}_t)\Big)\right] \\
&= \mathbb{E}_{t,c,\boldsymbol{x}_t}\left[\omega(t)\Big(\log q_\theta(\boldsymbol{x}_t|c) - \log p_{\text{ER}}(y^c, \boldsymbol{x}_t)\Big)\right] \\
&= \text{constant} - \mathbb{E}_{t,c,\boldsymbol{x}_t}\left[\omega(t)\,r\big(y^c, \hat{x}_0(\boldsymbol{x}_t)\big)\right],
\end{aligned}
\tag{12}
$$

where the first term is a constant because the distribution $q_\theta(x_t|c)$ is typically a uniformly-distributed collection of $N$ particles (i.e., $q_\theta(x_t|c) = 1/N$). Serving as a measure of the joint distribution of prompts and images, the explicit reward loss not only enhances text alignment during 3D generation but also provides the flexibility to incorporate additional criteria, such as human preference Xu et al. (2023); Murray et al. (2012), photorealism Kirstain et al. (2023), and geometric consistency Ye et al. (2024).

**Unified KL Divergence Framework.** Collecting these components, we can unify all loss terms in the diffusion- or reward-based text-to-3D generation framework by defining three core KL-based terms:

$$
\begin{aligned}
\mathcal{L}_{\text{CDP}}(\theta) &= \mathbb{E}_{t,c}\left[\omega(t)D_{\text{KL}}\Big(q_\theta(\boldsymbol{x}_t|c)\,\big\|\,p(\boldsymbol{x}_t|y^c)\Big)\right], \\
\mathcal{L}_{\text{UDP}}(\theta) &= \mathbb{E}_{t,c}\left[\omega(t)D_{\text{KL}}\Big(q_\theta(\boldsymbol{x}_t|c)\,\big\|\,p(\boldsymbol{x}_t)\Big)\right], \\
\mathcal{L}_{\text{ER}}(\theta) &= \mathbb{E}_{t,c}\left[\omega(t)D_{\text{KL}}\Big(q_\theta(\boldsymbol{x}_t|c)\,\big\|\,p_{\text{ER}}(y^c, \boldsymbol{x}_t)\Big)\right].
\end{aligned}
\tag{13}
$$

Both SDS and reward-guided objectives are simply linear combinations of these divergences:

$$\mathcal{L}_{\text{SDS}} = (1+\gamma)\mathcal{L}_{\text{CDP}} - \gamma\mathcal{L}_{\text{UDP}},$$
$$\mathcal{L}_{\text{Reward}} = \mathcal{L}_{\text{SDS}} + \lambda\mathcal{L}_{\text{ER}}. \tag{14}$$

This unified view permits flexible tuning of the weights on each term (see Appendix), yielding higher-fidelity generations. However, both theory and experiments Luo et al. (2025); Zhou et al. (2024b;a) show that relying on the inherently asymmetric KL divergence ($D_{\text{KL}}(q|p) \neq D_{\text{KL}}(p|q)$) destabilizes training and induces mode-seeking, thereby constraining the diversity of generated 3D assets.

DreamFusion Poole et al. (2022)  Fantasia3D Chen et al. (2023)  ProlificDreamer Wang et al. (2024)  **Dive3D (Ours)**

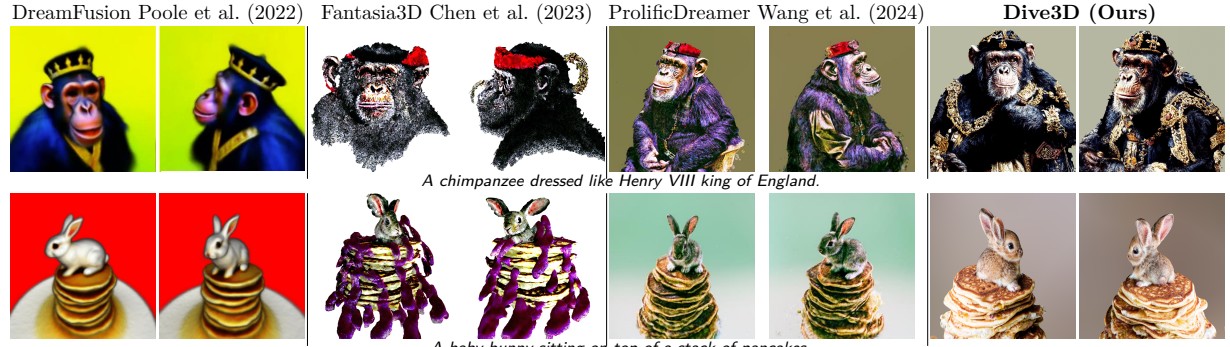

*A chimpanzee dressed like Henry VIII king of England.*

*A baby bunny sitting on top of a stack of pancakes.*

Figure 3: **Comparison with Baselines based on Stable Diffusion Rombach et al. (2022).** Dive3D exhibits higher quality, richer texture details, and superior alignment with human preferences, such as accurate clothing styles, and vivid fur texture.

MVDream Shi et al. (2023)  DreamReward Ye et al. (2024)  **Dive3D (Ours)**

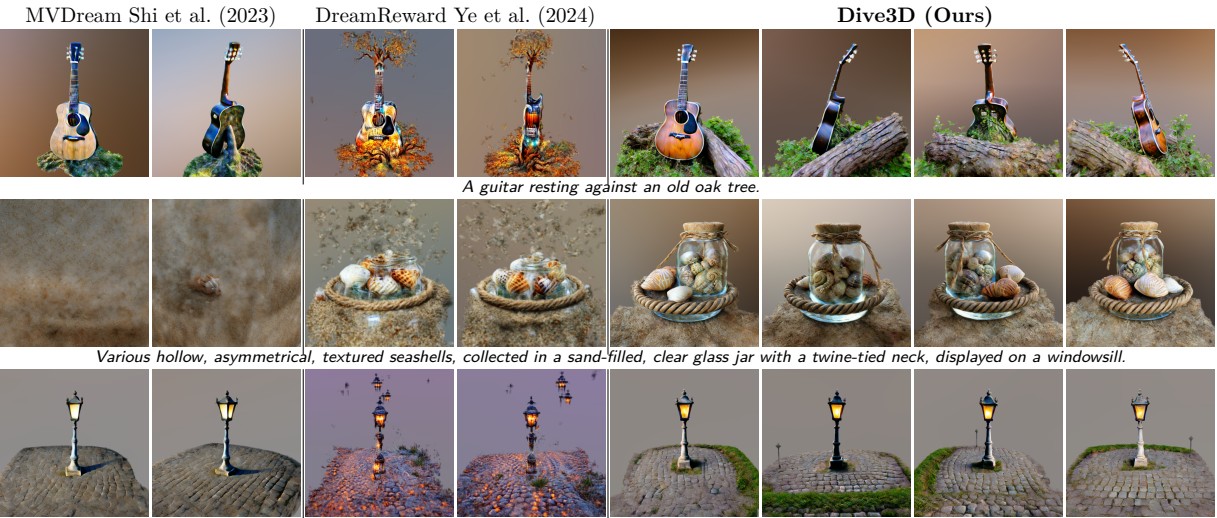

*A guitar resting against an old oak tree.*

*Various hollow, asymmetrical, textured seashells, collected in a sand-filled, clear glass jar with a twine-tied neck, displayed on a windowsill.*

*A sequence of street lamps, casting pools of light on cobblestone paths as twilight descends.*

Figure 4: **Comparison with Baselines based on MVDiffusion Shi et al. (2023) and reward model Ye et al. (2024).** Dive3D exhibits more detailed and realistic 3D generation, capturing fine-grained structures such as accurate guitar geometry and transparent glass materials.

## 4.2 From KL to Score-based Divergence

To mitigate these issues, in Dive3D we propose to replace the KL divergence with a score-based divergence, named score implict matching (SIM) loss(Luo et al., 2024b), which has shown significant improvements in generation diversity in one-step diffusion and flow models(Luo et al., 2024b; Zhou et al., 2024b; Huang et al., 2024b). Specifically, the score-based divergence is defined between two distributions $p$ and $q$ as

$$D_{[0,T]}(p,q) = \int_0^T w(t)\, \mathbb{E}_{\boldsymbol{x}_t \sim \pi_t}\Big[d\big(s_p(\boldsymbol{x}_t) - s_q(\boldsymbol{x}_t)\big)\Big]\, dt, \tag{15}$$

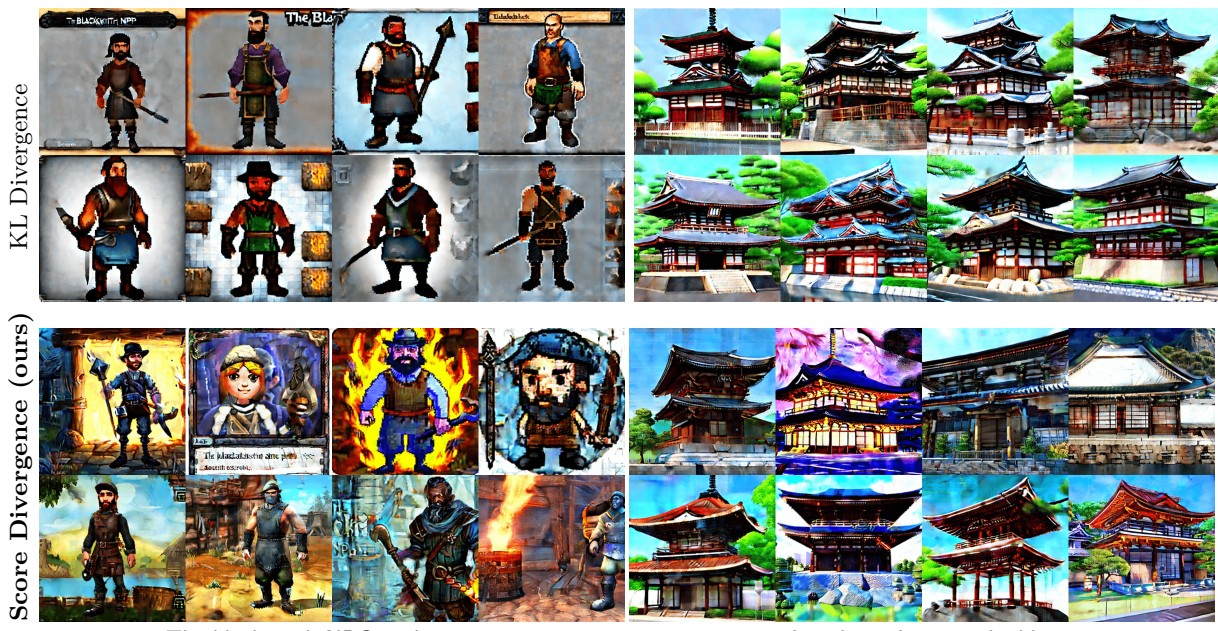

Figure 5: **Score-based divergence vs. KL divergence in 2D space sampling.** The proposed score-based divergence significantly enhances the diversity of generated 2D samples, yielding more varied backgrounds and clothing in "game character" generation, as well as a broader range of environments, lighting conditions, and architectural features in "Japanese building" generation.

where the score functions of these two distributions are given by $s_p(\boldsymbol{x}_t) = \nabla_{\boldsymbol{x}_t} \log p(\boldsymbol{x}_t)$ and $s_q(\boldsymbol{x}_t) = \nabla_{\boldsymbol{x}_t} \log q(\boldsymbol{x}_t)$, $d : \mathbb{R}^d \to \mathbb{R}$ is a distance function, $\pi_t$ is a sampling distribution whose support exceeds that of $p_t$ and $q_t$, and $w(t)$ is a weighting function. If we set $p(\cdot) = p(\boldsymbol{x}_t|y^c), p(\boldsymbol{x}_t), p_{\mathrm{ER}}(y^c, \boldsymbol{x}_t)$ and $q(\cdot) = q_\theta(\boldsymbol{x}_t|c)$, then the KL-based losses in Eqs. 13-14 can be updated to

$$
\begin{aligned}
\mathcal{L}_{\mathrm{Score-CDP}}(\theta) &= \int_0^T w(t)\, \mathbb{E}_{\boldsymbol{x}_t \sim \pi_t}\Big[d\Big(s_p(\boldsymbol{x}_t|y^c) - s_{q_\theta}(\boldsymbol{x}_t|c)\Big)\Big]\, dt, \\
\mathcal{L}_{\mathrm{Score-UDP}}(\theta) &= \int_0^T w(t)\, \mathbb{E}_{\boldsymbol{x}_t \sim \pi_t}\Big[d\Big(s_p(\boldsymbol{x}_t) - s_{q_\theta}(\boldsymbol{x}_t|c)\Big)\Big]\, dt, \\
\mathcal{L}_{\mathrm{Score-ER}}(\theta) &= \int_0^T w(t)\, \mathbb{E}_{\boldsymbol{x}_t \sim \pi_t}\Big[d\Big(\nabla_{\boldsymbol{x}_t} r\big(y^c, \hat{x}_0(\boldsymbol{x}_t)\big) - s_{q_\theta}(\boldsymbol{x}_t|c)\Big)\Big]\, dt, \\
\mathcal{L}_{\mathrm{Dive3D}} &= (1+\gamma)\mathcal{L}_{\mathrm{Score-CDP}} - \gamma\mathcal{L}_{\mathrm{Score-UDP}} + \lambda\mathcal{L}_{\mathrm{Score-ER}}
\end{aligned}
\tag{16}
$$

This formulation offers a more effective similarity metric between the generated content and diffusion- or reward-based image distributions, yielding 3D outputs that are both more diverse and higher fidelity than those produced using traditional KL divergence.

Although this divergence may initially seem intractable, recent work (Luo et al., 2025) shows that the gradient of this divergence with respect to $\theta$ can be efficiently computed without directly differentiating the score functions by introducing a separate approximation network. For a full derivation and implementation details, please refer to Appendix.

### 4.3 Analysis of Score-based Divergence

Our methodology, which leverages Score-based divergence (SIM), offers significant advantages over conventional generative objectives based on KL-divergence, such as those used in SDS. The primary limitation of KL-divergence-based methods is their inherent "mode-seeking" nature, which frequently leads to mode collapse. This results in generated samples that, while often high in quality, lack diversity.

This phenomenon can be intuitively understood by analogizing the target data distribution to a complex mountain range with multiple peaks, where each peak represents a data mode. A KL-based objective function akin to a GPS system that identifies the most efficient path to the single highest peak, consequently ignoring all other valid destinations. In contrast, our SIM-based approach constructs a complete topographical map of the entire landscape. By matching the score function—the gradient of the log-probability density—at every point, our model illuminates the complete topology of the data distribution. This enables the generation of truly diverse results that cover all modes of the distribution.

From a theoretical standpoint, the mode collapse issue in the KL-divergence stems from its mathematical formulation, which seeks to minimize the following objective:

$$D_{KL}(P||Q) = \mathbb{E}_{x \sim P} \log \frac{P(x)}{Q(x)} \tag{17}$$

The objective's reliance on the likelihood ratio $P(x)/Q(x)$ renders it ill-defined and unstable, particularly when the support of the generated distribution ($P$) and the target distribution ($Q$) are misaligned. This instability compels the model to converge onto a single, "safe" mode of $Q$ where the ratio is well-behaved, thereby discarding other modes.

Our SIM-based approach circumvents this fundamental flaw by minimizing a score-based divergence instead:

$$D_{SIM}(P||Q) = \mathbb{E}_{x \sim Q} d[\nabla_x \log P(x) - \nabla_x \log Q(x)] \tag{18}$$

Rather than comparing an unstable ratio of densities, this objective directly aligns the vector fields of the score functions ($\nabla_x \log P(x)$ and $\nabla_x \log Q(x)$). This score-matching process is inherently robust to misaligned support between distributions. Consequently, our model is empowered to learn the entire topological landscape of the target distribution, thereby preserving all of its modes and fostering superior generative diversity.

## 5 Experiment

In this section, we evaluate how our proposed score-based divergence optimization enhances both quality and diversity in text-to-3D synthesis. We perform comprehensive experiments on the GPTEval3D benchmark Wu et al. (2024a), supplemented by additional 2D and 3D assessments that demonstrate the effectiveness and diversity of the method.

### 5.1 Evaluation on the GPTEval3D Benchmark

**Setup.** We first evaluate Dive3D on 110 creative and complex prompts from the GPTEval3D benchmark Wu et al. (2024a), comparing against 9 state-of-the-art methods, including DreamFusion Poole et al. (2022), DreamGaussian Tang et al. (2023b), Instant3D Li et al. (2023), Fantasia3D Chen et al. (2023), Latent-NeRF Metzer et al. (2022), Magic3D Lin et al. (2023), ProlificDreamer Wang et al. (2023c), MVDream Shi et al. (2023), and DreamReward Ye et al. (2024). All experiments use PyTorch and the ThreeStudio framework Guo et al. (2023), testing both MVDream Shi et al. (2023) and Stable Diffusion Rombach et al. (2022) as diffusion backbones, and PickScore Kirstain et al. (2023) as the reward model. Optimization takes about one hour per object on a single NVIDIA A100 GPU.

**Quantitative Results.** Table 1 reports performance of our method across six metrics, including text-asset alignment (+53.5), 3D plausibility (+49), text-geometry alignment (+68.2), texture details (+67.5), geometry details (+35.3), and overall performance (+50.0), where "+" indicates improvement and "–" indicates degradation relative to the state of the art. Dive3D achieves the top rank on every metric, demonstrating that score-based divergence guidance—especially when combined with reward models—yields substantial gains over both diffusion-only and reward-augmented baselines.

**Qualative Results.** Figure 3 compares Dive3D against methods built on Stable Diffusion (e.g., Dream-Fusion, Fantasia3D, ProlificDreamer), which often struggle with fine details or prompt adherence. By optimizing a score-based divergence that unifies text-conditioned diffusion priors with a differentiable reward model, Dive3D consistently produces high-fidelity, semantically precise 3D assets.

Table 1: **Quantitative Results on 110 Prompts from the GPTEval3D Benchmark Wu et al. (2024a).** We compute all six GPTEval3D metrics—text alignment, 3D plausibility, texture–geometry coherence, geometry details, texture details, and overall score—to comprehensively evaluate 3D generation quality. Dive3D achieves the highest score on every metric, demonstrating its superior performance.

| Method | Prompts from GPTEval3D Wu et al. (2024a) | | | | | |
|---|---|---|---|---|---|---|
| | Alignment | Plausibility | T-G Coherency. | Geo Details | Tex Details | Overall |
| DreamFusionPoole et al. (2022) | 1000.0 | 1000.0 | 1000.0 | 1000.0 | 1000.0 | 1000.0 |
| DreamGaussianTang et al. (2023b) | 1100.6 | 953.6 | 1158.6 | 1126.2 | 1130.8 | 951.4 |
| Fantasia3DChen et al. (2023) | 1067.9 | 891.9 | 1006.0 | 1109.3 | 1027.5 | 933.5 |
| Instant3DLi et al. (2023) | 1200.0 | 1087.6 | 1152.7 | 1152.0 | 1181.3 | 1097.8 |
| Latent-NeRFMetzer et al. (2022) | 1222.3 | 1144.8 | 1156.7 | 1180.5 | 1160.8 | 1178.7 |
| Magic3DLin et al. (2023) | 1152.3 | 1000.8 | 1084.4 | 1178.1 | 1084.6 | 961.7 |
| ProlificDreamerWang et al. (2023c) | 1261.8 | 1058.7 | 1152.0 | 1246.4 | 1180.6 | 1012.5 |
| SyncDreamerLiu et al. (2024) | 1041.2 | 968.8 | 1083.1 | 1064.2 | 1045.7 | 963.5 |
| MVDreamShi et al. (2023) | 1270.5 | 1147.5 | 1250.6 | 1324.9 | 1255.5 | 1097.7 |
| DreamReward[1]Ye et al. (2024) | 1287.5 | 1195.0 | 1254.4 | 1295.5 | 1261.6 | 1193.3 |
| DIVE3D (Ours) | **1341.0** | **1249.0** | **1322.6** | **1360.2** | **1329.1** | **1243.3** |

[1] Our metrics differ from those reported in the original DreamReward paper because GPT-4V has been deprecated in GPTEval3D, so we instead use GPT-4o-mini.

Additional examples in Figures 4 and 10 compare Dive3D with MVDream and DreamReward. While MVDream preserves geometric consistency, it sometimes deviates from the prompt content (missing keywords highlighted in red). DreamReward improves alignment but remains constrained by its KL-based formulation and associated mode collapse. In contrast, Dive3D faithfully follows the prompt, delivers rich detail and appealing aesthetics, and maintains strong visual coherence.

Table 2: Quantitative comparison of diversity and quality. Arrows indicate whether higher ($\uparrow$) or lower ($\downarrow$) values are better.

| Method | CLIP-D $\uparrow$ | LPIPS-D $\uparrow$ | Chamfer-D $\uparrow$ | FID $\downarrow$ |
|---|---|---|---|---|
| Prolific Dreamer (KL-based) | 0.3908 | 0.485 | 0.072 | 207.34 |
| **Dive3D (Ours, Score-based)** | **0.4483** | **0.551** | **0.115** | **168.59** |

## 5.2 Analysis on Generation Diversity

**Setup.** We then show that score-based divergences produce more diverse, information-rich outputs than traditional KL-based losses. To evaluate this, we test our method in both 2D and 3D settings—using Stable Diffusion Rombach et al. (2022) as the backbone. In 2D, we represent scenes with 2D Neural Radiance Fields; in 3D, we use full 3D NeRFs. We primarily compare against ProlificDreamer Wang et al. (2023c), the leading KL-divergence–based method that leverages variational score distillation (VSD) to maximize diversity in text-to-3D generation. On a single NVIDIA A100 GPU, our 2D experiments complete in roughly 30 minutes, while the 3D evaluations take about 9 hours.

**2D Results.** We begin by evaluating 2D generation, where we distill a 2D neural field from a text-to-image diffusion model. This task shares the same mathematical formulation as our text-to-3D problem but is computationally less demanding because it does not involve camera poses. As shown in Fig. 5, for both game character and realistic architecture generation tasks, the score-based divergence consistently produces more diverse samples than KL divergence. For instance, when generating "a realistic Japanese building," the KL-based method consistently generates towers with standard color schemes (predominantly red and blue), uniform backgrounds (lush green trees), and similar weather and time conditions (sunny daytime). In contrast, the score-based approach generates outputs with varied lighting (e.g., night scenes, snowy settings) and diverse architectural features (e.g., towers, pavilions, and residential houses). A similar trend is observed

in the game character generation task: while the KL-based SDS loss tends to produce similar archetypes, the score-based loss reveals a wider range of characters, clothing styles, and backgrounds.

**3D Results.** These diversity gains naturally and effectively generalize to 3D synthesis. Figure 1(a) compares the output for "a pirate ship in the sky" under the KL-based VSD loss versus our score-based divergence. As expected, our approach produces a far wider range of geometric shapes, surface textures, and background scenes—from bright sunny skies to dark thunderous clouds. Figure 11 offers additional examples across diverse prompts to reinforce this finding, illustrating how score-based divergence yields richer variation in colors, object styles, material properties, and environmental details.

To quantitatively validate that our method enhances diversity, we have also conducted a large-scale evaluation. We generated 12,800 images from 50 diverse prompts and assessed them using three standard metrics: Pairwise CLIP-Distance (CLIP-D) for inter-sample diversity, average pairwise LPIPS for perceptual diversity, and Fréchet Inception Distance (FID) to evaluate overall set quality. As summarized in Table 2, our method significantly outperforms the KL-based baseline (ProlificDreamer) on all three metrics. These improvements are statistically significant, as all $p < 0.01$ with paired t-tests ($t_{CLIP}(49) = 5.21$ and $t_{FID}(49) = -8.83$), providing strong quantitative evidence that our method effectively mitigates mode collapse.

## 5.3 Computational Cost

We analyze the runtime and memory overhead of Dive3D in Table 3. The cost depends on the diffusion backbone. When using MVDiffusion, our method is comparable to other reward-guided methods like DreamReward. When using Stable Diffusion, our method is slightly slower than ProlificDreamer (NeRF) and requires more memory due to the reward guidance. In both scenarios, the computational cost is justified by the significant gains in generation quality and diversity. All experiments were run on a single NVIDIA A100 GPU.

Table 3: Computational cost and performance comparison, broken down by diffusion backbone.

| Diffusion Backbone | Method | Generation Time | Peak Memory (GB) |
|---|---|---|---|
| MVDiffusion | MVDream | 0.7 hours | ∼14 |
| | DreamReward | 1.0 hours | ∼22 |
| | **Dive3D (Ours)** | **1.0 hours** | **∼22** |
| Stable Diffusion | ProlificDreamer | 7.9 hours | ∼27 |
| | **Dive3D (Ours)** | **9.0 hours** | **∼35** |

## 6 Conclusion

In this paper, we present Dive3D, a novel framework that boosts both diffusion-based distillation and reward-guided optimization by substituting their asymmetric KL-divergence objectives with a powerful score-based divergence. Evaluated on the GPTEval3D and other benchmarks, Dive3D effectively alleviates mode collapse, yielding substantially greater diversity while simultaneously improving text alignment, geometric plausibility, and visual fidelity.

**Limitations and Future Work.** Although Dive3D delivers compelling results, its runtime remains slower than recent LRM-based methods. In future work, we plan to integrate our score-based divergence with latent reconstruction models by first distilling diverse, text-driven multi-view generators and then combining them with LRM techniques to achieve rapid, high-fidelity, and diverse 3D synthesis.

## ACKNOWLEDGMENTS

This work was supported by the National Natural Science Foundation of China(62371007), the National Key Research and Development Program of China (2024YFC3406400), and Biomedical Computing Platform of National Biomedical Imaging Center, Peking University.

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

# A  Derivation of the Score-based Loss Function

The derivation is based on the so-called score-projection identity which was first found in Vincent (2011) to bridge denoising score matching and denoising auto-encoders. Later, the identity is applied by Zhou et al. (2024b); Luo et al. (2025) for deriving distillation methods for image generation. We appreciate the efforts of Zhou et al. (2024b) and introduce the conditional score-projection identity for inverse problems here:

**Theorem A.1.** *(Conditional score-projection identity). Let $\boldsymbol{u}(\mathring{\mathrm{u}}, \theta)$ be a vector-valued function, under mild conditions, the identity holds:*

$$\mathbb{E}_{\substack{\boldsymbol{x}_0 \sim p_\theta(\boldsymbol{x}_0|\boldsymbol{y}) \\ \boldsymbol{x}_t|\boldsymbol{x}_0 \sim q_t(\boldsymbol{x}_t|\boldsymbol{x}_0)}} \boldsymbol{u}(\boldsymbol{x}_t, \boldsymbol{y}, \theta)^T \left\{ \boldsymbol{s}_{p_{\theta,t}}(\boldsymbol{x}_t|\boldsymbol{y}) - \nabla_{\boldsymbol{x}_t} \log q_t(\boldsymbol{x}_t|\boldsymbol{x}_0) \right\} = 0, \quad \forall \theta. \tag{19}$$

Next, we turn to derive the loss function.

*Proof.* We prove a more general result. Let $\boldsymbol{u}(\mathring{\mathrm{u}})$ be a vector-valued function, the so-called score-projection identity Vincent (2011) holds,

$$\mathbb{E}_{\substack{\boldsymbol{x}_0 \sim p_\theta(\boldsymbol{x}_0|\boldsymbol{y}) \\ \boldsymbol{x}_t|\boldsymbol{x}_0 \sim q_t(\boldsymbol{x}_t|\boldsymbol{x}_0)}} \boldsymbol{u}(\boldsymbol{x}_t, \boldsymbol{y}, \theta)^T \left\{ \boldsymbol{s}_{p_{\theta,t}}(\boldsymbol{x}_t|\boldsymbol{y}) - \nabla_{\boldsymbol{x}_t} \log q_t(\boldsymbol{x}_t|\boldsymbol{x}_0) \right\} = 0, \quad \forall \theta. \tag{20}$$

Taking $\theta$ gradient on both sides of identity 20, we have:

$$
\begin{aligned}
0 &= \mathbb{E}_{\substack{\boldsymbol{x}_0 \sim p_\theta(\boldsymbol{x}_0|\boldsymbol{y}) \\ \boldsymbol{x}_t|\boldsymbol{x}_0 \sim q_t(\boldsymbol{x}_t|\boldsymbol{x}_0)}} \frac{\partial}{\partial \boldsymbol{x}_t} \left\{ \boldsymbol{u}(\boldsymbol{x}_t, \boldsymbol{y}, \theta)^T \left\{ \boldsymbol{s}_{p_{\theta,t}}(\boldsymbol{x}_t|\boldsymbol{y}) - \nabla_{\boldsymbol{x}_t} \log q_t(\boldsymbol{x}_t \mid \boldsymbol{x}_0) \right\} \right\} \frac{\partial \boldsymbol{x}_t}{\partial \theta} \\
&+ \mathbb{E}_{\substack{\boldsymbol{x}_0 \sim p_\theta(\boldsymbol{x}_0|\boldsymbol{y}) \\ \boldsymbol{x}_t|\boldsymbol{x}_0 \sim q_t(\boldsymbol{x}_t|\boldsymbol{x}_0)}} \frac{\partial}{\partial \boldsymbol{x}_0} \left\{ \boldsymbol{u}(\boldsymbol{x}_t, \boldsymbol{y}, \theta)^T \left\{ -\nabla_{\boldsymbol{x}_t} \log q_t(\boldsymbol{x}_t \mid \boldsymbol{x}_0) \right\} \right\} \frac{\partial \boldsymbol{x}_0}{\partial \theta} \\
&+ \mathbb{E}_{\substack{\boldsymbol{x}_0 \sim p_\theta(\boldsymbol{x}_0|\boldsymbol{y}) \\ \boldsymbol{x}_t|\boldsymbol{x}_0 \sim q_t(\boldsymbol{x}_t|\boldsymbol{x}_0)}} \boldsymbol{u}(\boldsymbol{x}_t, \boldsymbol{y}, \theta)^T \frac{\partial}{\partial \theta} \left\{ \boldsymbol{s}_{p_{\theta,t}}(\boldsymbol{x}_t|\boldsymbol{y}) \right\} + \frac{\partial}{\partial \theta} \boldsymbol{u}(\boldsymbol{x}_t, \boldsymbol{y}, \theta)^T \boldsymbol{s}_{p_{\theta,t}}(\boldsymbol{x}_t|\boldsymbol{y}) \\
&= \mathbb{E}_{\substack{\boldsymbol{x}_0 \sim p_\theta(\boldsymbol{x}_0|\boldsymbol{y}) \\ \boldsymbol{x}_t|\boldsymbol{x}_0 \sim q_t(\boldsymbol{x}_t|\boldsymbol{x}_0)}} \boldsymbol{u}(\boldsymbol{x}_t, \boldsymbol{y}, \theta)^T \frac{\partial}{\partial \theta} \left\{ \boldsymbol{s}_{p_{\theta,t}}(\boldsymbol{x}_t|\boldsymbol{y}) \right\} \\
&+ \mathbb{E}_{\substack{\boldsymbol{x}_0 \sim p_\theta(\boldsymbol{x}_0|\boldsymbol{y}) \\ \boldsymbol{x}_t|\boldsymbol{x}_0 \sim q_t(\boldsymbol{x}_t|\boldsymbol{x}_0)}} \left\{ \frac{\partial}{\partial \boldsymbol{x}_t} \left\{ \boldsymbol{u}(\boldsymbol{x}_t, \boldsymbol{y}, \theta)^T \left\{ \boldsymbol{s}_{p_{\theta,t}}(\boldsymbol{x}_t|\boldsymbol{y}) - \nabla_{\boldsymbol{x}_t} \log q_t(\boldsymbol{x}_t|\boldsymbol{x}_0) \right\} \right\} \frac{\partial \boldsymbol{x}_t}{\partial \theta} \right. \\
&+ \left. \frac{\partial}{\partial \boldsymbol{x}_0} \left\{ \boldsymbol{u}(\boldsymbol{x}_t, \boldsymbol{y}, \theta)^T \left\{ -\nabla_{\boldsymbol{x}_t} \log q_t(\boldsymbol{x}_t|\boldsymbol{x}_0) \right\} \right\} \frac{\partial \boldsymbol{x}_0}{\partial \theta} + \frac{\partial}{\partial \theta} \boldsymbol{u}(\boldsymbol{x}_t, \boldsymbol{y}, \theta)^T \boldsymbol{s}_{p_{\theta,t}}(\boldsymbol{x}_t|\boldsymbol{y}) \right\} \\
&= \mathbb{E}_{\boldsymbol{x}_t \sim p_{\theta,t}} \boldsymbol{u}(\boldsymbol{x}_t, \boldsymbol{y}, \theta)^T \frac{\partial}{\partial \theta} \left\{ \boldsymbol{s}_{p_{\theta,t}}(\boldsymbol{x}_t|\boldsymbol{y}) \right\} + \frac{\partial}{\partial \theta} \mathbb{E}_{\boldsymbol{x}_0 \sim p_{\theta,t}} \boldsymbol{u}(\boldsymbol{x}_t, \boldsymbol{y}, \theta)^T \left\{ \boldsymbol{s}_{p_{\mathrm{sg}[\theta],t}}(\boldsymbol{x}_t|\boldsymbol{y}) - \nabla_{\boldsymbol{x}_t} \log q_t(\boldsymbol{x}_t|\boldsymbol{x}_0) \right\}
\end{aligned}
\tag{21}
$$

Therefore, we have the following identity:

$$\mathbb{E}_{\boldsymbol{x}_t \sim p_{\theta,t}} \boldsymbol{u}(\boldsymbol{x}_t, \boldsymbol{y}, \theta)^T \frac{\partial}{\partial \theta} \left\{ \boldsymbol{s}_{p_{\theta,t}}(\boldsymbol{x}_t|\boldsymbol{y}) \right\} = -\frac{\partial}{\partial \theta} \mathbb{E}_{\boldsymbol{x}_0 \sim p_{\theta,t}} \boldsymbol{u}(\boldsymbol{x}_t, \boldsymbol{y}, \theta)^T \left\{ \boldsymbol{s}_{p_{\mathrm{sg}[\theta],t}}(\boldsymbol{x}_t|\boldsymbol{y}) - \nabla_{\boldsymbol{x}_t} \log q_t(\boldsymbol{x}_t|\boldsymbol{x}_0) \right\} \tag{22}$$

which holds for arbitrary function $\boldsymbol{u}(\mathring{\mathrm{u}}, \theta)$ and parameter $\theta$. If we set

$$
\begin{aligned}
\boldsymbol{u}(\boldsymbol{x}_t, \boldsymbol{y}, \theta) &= \mathbf{d}'(\boldsymbol{k}_t) \\
\boldsymbol{k}_t &= \boldsymbol{s}_{p_{\mathrm{sg}[\theta],t}}(\boldsymbol{x}_t|\boldsymbol{y}) - s_{q_t}(\boldsymbol{x}_t) - s_{q_t}(\boldsymbol{y}|\boldsymbol{x}_t)
\end{aligned}
\tag{23}
$$

Then we formally have the loss function as:

$$loss = \mathbb{E}_{\boldsymbol{x}_t \sim p_{\theta,t}} \left\{ -\mathbf{d}'(\boldsymbol{k}_t) \right\}^T \left\{ \boldsymbol{s}_{p_{\mathrm{sg}[\theta],t}}(\boldsymbol{x}_t|\boldsymbol{y}) - \nabla_{\boldsymbol{x}_t} \log q_t(\boldsymbol{x}_t|\boldsymbol{x}_0) \right\} \tag{24}$$

# B    Theoretical Proof of SIM's Robustness Against Mode Collapse

This section provides a formal mathematical proof that the score-based loss (SIM) function is theoretically more robust against mode collapse than the KL divergence-based loss function. The proof is structured by first formally defining a mode-seeking loss function, then proving that KL-divergence satisfies this definition while SIM does not. This culminates in the main theorem establishing SIM's theoretical advantage.

## B.1    Definitions

**Definition B.1** (Mode Collapse). Let $Q(x)$ be a target probability distribution on a space $\mathcal{X} \subseteq \mathbb{R}^d$, possessing a set of distinct modes (high-density regions) $\{\mathcal{M}_i\}_{i=1}^k$. Let $P_\theta(x)$ be the distribution generated by a model with parameters $\theta$. The model is said to suffer from *mode collapse* if the support of its distribution, $\mathrm{supp}(P_\theta)$, fails to cover the union of the significant modes of $Q$, i.e.,

$$\mathrm{supp}(P_\theta) \subsetneq \bigcup_{i=1}^k \mathcal{M}_i. \tag{25}$$

**Definition B.2** (Mode-Seeking Loss Function). A loss function $L(P_\theta, Q)$ is defined as **mode-seeking** if it satisfies the following property: For any region $\mathcal{R} \subset \mathcal{X}$ of extremely low target density (i.e., $\sup_{x \in \mathcal{R}} Q(x) < \epsilon$ for some $\epsilon \to 0$), if the generated distribution assigns a non-vanishing probability mass to this region (i.e., $\int_\mathcal{R} P_\theta(x)dx > c$ for some constant $c > 0$), then the loss must diverge,

$$L(P_\theta, Q) \to \infty. \tag{26}$$

Intuitively, a mode-seeking loss function infinitely penalizes the generator for exploring low-density regions of the target distribution.

## B.2    Lemmas

**Lemma B.1.** *The Kullback-Leibler (KL) divergence, $D_{KL}(P_\theta \| Q)$, is a mode-seeking loss function.*

*Proof.* The KL-divergence is defined as:

$$L_{KL}(P_\theta) = D_{KL}(P_\theta \| Q) = \int_\mathcal{X} P_\theta(x) \log \frac{P_\theta(x)}{Q(x)} dx \tag{27}$$

$$= \int_\mathcal{X} P_\theta(x) \log P_\theta(x) dx - \int_\mathcal{X} P_\theta(x) \log Q(x) dx. \tag{28}$$

Let us analyze the second term, the negative cross-entropy $T = - \int_\mathcal{X} P_\theta(x) \log Q(x) dx$.

Consider a region $\mathcal{R}$ that satisfies the condition in Definition 2, such that $\forall x \in \mathcal{R}, Q(x) < \epsilon$ with $\epsilon \to 0$. As a consequence, $\log Q(x) \to -\infty$ and thus $-\log Q(x) \to +\infty$ for all $x \in \mathcal{R}$.

Now, assume the generator assigns a non-vanishing probability mass to this region, $\int_\mathcal{R} P_\theta(x)dx > c > 0$. The contribution to the loss from this region is:

$$T_\mathcal{R} = - \int_\mathcal{R} P_\theta(x) \log Q(x) dx. \tag{29}$$

Since $P_\theta(x)$ is a probability density function and its integral over $\mathcal{R}$ is a positive constant, while $-\log Q(x)$ diverges to $+\infty$ over this entire region, the value of the integral must also diverge:

$$T_\mathcal{R} \to \infty. \tag{30}$$

As the total loss $L_{KL}(P_\theta)$ is the sum of terms including $T_\mathcal{R}$, it follows that $L_{KL}(P_\theta) \to \infty$. Thus, $D_{KL}(P_\theta \| Q)$ satisfies the property of a mode-seeking loss function as per Definition 2. □

**Lemma B.2.** *The SIM loss function, $L_{SIM}$, is not a mode-seeking loss function, under the mild assumption that $Q(x)$ is continuously differentiable.*

*Proof.* The SIM loss is defined by matching the score functions of the two distributions, $s_P(x) = \nabla_x \log P(x)$ and $s_Q(x) = \nabla_x \log Q(x)$:

$$L_{SIM}(P_\theta) = \mathbb{E}_{x \sim \pi(x)} \left[ \|\nabla_x \log P_\theta(x) - \nabla_x \log Q(x)\|^2 \right], \tag{31}$$

where $\pi(x)$ is a sampling distribution. We assume that the target distribution $Q(x)$ is continuously differentiable ($C^1$ smooth), which is a standard assumption for distributions defined by stochastic differential equations in diffusion models.

Consider again the region $\mathcal{R}$ where $\sup_{x \in \mathcal{R}} Q(x) \to 0$. The critical term to analyze is the target score function, $s_Q(x) = \nabla_x \log Q(x)$. By the chain rule, this is $s_Q(x) = \frac{\nabla_x Q(x)}{Q(x)}$. Although the denominator $Q(x) \to 0$, the numerator $\nabla_x Q(x)$ also approaches zero if $x$ is in a region of a local minimum of the probability density. For smooth distributions like Gaussian mixtures, the score function is well-defined and bounded across the entire space. Even in low-density regions between modes, the score vector $\nabla_x \log Q(x)$ is a finite vector pointing towards the nearest mode; it does not diverge to infinity.

Therefore, on any compact subset of $\mathcal{X}$, the score function $s_Q(x)$ is bounded, i.e., there exists a constant $K$ such that $\|s_Q(x)\| < K$. Consequently, the loss contribution from any point $x \in \mathcal{R}$ is:

$$\|\nabla_x \log P_\theta(x) - s_Q(x)\|^2, \tag{32}$$

which remains a finite value. The loss $L_{SIM}$ does not diverge to infinity merely because the generator samples from a low-density region of the target. Thus, $L_{SIM}$ does not satisfy the property of a mode-seeking loss function as per Definition 2. $\square$

## B.3 Main Theorem

**Theorem B.3.** *An optimization process guided by a mode-seeking loss function is theoretically susceptible to mode collapse. Conversely, a process guided by a loss function that is not mode-seeking is theoretically more robust against mode collapse.*

*Proof.* From Lemma 1, we have proven that the KL-divergence is a mode-seeking loss function. Any optimization algorithm (e.g., gradient descent) attempting to minimize $L_{KL}$ will receive an unbounded penalty for placing probability mass outside the modes of $Q$. The optimizer is therefore strongly incentivized to adjust $\theta$ such that the support of $P_\theta$ becomes a strict subset of the high-density regions of $Q$. This directly leads to the condition of mode collapse as defined in Definition 1.

From Lemma 2, we have proven that the SIM loss is not mode-seeking. An optimizer minimizing $L_{SIM}$ receives finite and meaningful gradient information across the entire data space, including low-density regions. There is no infinite penalty that forces the optimizer to "escape" these regions. Instead, it is guided to match the vector field of score functions globally. This allows the learned distribution $P_\theta$ to capture the full topology of the target distribution $Q$, making the optimization process inherently more robust to mode collapse.

Therefore, the SIM framework provides a stronger theoretical foundation for generating diverse samples compared to frameworks based on KL-divergence. $\square$

## C Implementation Details

**Pseudo-code for Dive3D**   A detailed pseudo-code for Dive3D is presented in algorithm 1.

In this paper, we conduct experiments primarily using a NVIDIA A100 Tensor Core GPU, with the former mainly employed for NeRF and 3D Gaussian Splatting generation. For our quantitative evaluations, we maintain consistent experimental setups across all methods, employing similar pipelines that include 3D

---

**Algorithm 1** Pseudo-code for Dive3D

---

1: **Input:** Text-to-image diffusion score network $s_\phi(\cdot,\cdot,\cdot)$ and its LoRA parameters $\varphi$, Reward function $R(\cdot,\cdot)$
2: **Input:** Text prompt $y$, Differentiable 3D renderer $g(\theta, v)$
3: **Input:** Learning rate $\eta$ for 3D parameters $\theta$, Timestep weighting function $w(t)$
4: **Input:** Hyperparameters: $\gamma$ (guidance scale coefficient), $\lambda$ (reward weight)
5: **Input:** Noise schedule parameters $\alpha_t, \sigma_t$
6:
7: **Initialization:** Randomly initialize 3D representation parameters $\theta_0$
8:
9: **while** not converged **do**
10:      Sample camera viewpoint $v$
11:      Render 2D image from current 3D model: $x_0 = g(\theta, v)$
12:      Sample timestep $t \sim \text{Uniform}(0, T)$ and noise $\epsilon \sim \mathcal{N}(0, I)$
13:      Compute noisy image: $x_t = \alpha_t x_0 + \sigma_t \epsilon$
14:      Compute score of the current rendered data distribution: $s_{rendered} = -\epsilon/\sigma_t$
15:
16:      **Score-based Conditional Diffusion Prior (S-CDP) Gradient**
17:          Estimate conditional target score: $s_{cond\_target} = s_\phi(x_t, y, t)$
18:          Compute S-CDP gradient: $\nabla_\theta \mathcal{L}_{S-CDP} \approx w(t) d_{score}(s_{cond\_target}, s_\varphi(x_t, y, t))$
19:
20:      **Score-based Unconditional Diffusion Prior (S-UDP) Gradient**
21:          Estimate unconditional target score: $s_{uncond\_target} = s_\phi(x_t, \emptyset, t)$ (using null text $\emptyset$)
22:          Compute S-UDP gradient: $\nabla_\theta \mathcal{L}_{S-UDP} \approx w(t) d_{score}(s_{uncond\_target}, s_\varphi(x_t, y, t))$
23:
24:      **Score-based Explicit Reward (S-ER) Gradient**
25:          Estimate denoised image $\hat{x}_0(x_t)$ (e.g., using $x_t, s_\phi, \alpha_t, \sigma_t$)
26:          Estimate target reward score: $s_{reward\_target} = \nabla_{x_t} R(y, \hat{x}_0(x_t))$
27:          Compute S-ER gradient: $\nabla_\theta \mathcal{L}_{S-ER} \approx w(t) d_{score}(s_{reward\_target}, s_\varphi(x_t, y, t))$
28:
29:      **Total Loss Gradient and Parameter Update**
30:          Compute total Dive3D loss gradient (based on Eq. 16):
31:          $\nabla_\theta \mathcal{L}_{Dive3D} = (1 + \gamma)\nabla_\theta \mathcal{L}_{S-CDP} - \gamma \nabla_\theta \mathcal{L}_{S-UDP} + \lambda \nabla_\theta \mathcal{L}_{S-ER}$
32:          Update parameters: $\theta \leftarrow \theta - \eta \nabla_\theta \mathcal{L}_{Dive3D}$
33:          Update parameters: $\varphi \leftarrow DenoisingScoreMatching(x_t, t)$
34: **end while**

---

representation selection, training procedures, shape initialization, and the use of a teacher diffusion model. We uniformly apply identical negative prompts during the assessments. The primary distinction between the methods lies in their respective loss functions. In our experiments, we use a CFG scale of 7.5 for mesh generation and 20 for Gaussian Splatting generation. We observed that a high CFG scale tends to oversaturate colors, while a low CFG scale hinders object convergence. With algorithmic improvements, we found that a smaller CFG scale can produce high-quality 3D objects with smooth colors.

For text-to-3D generation, we utilize Stable Diffusion v2.1 and the MVDream model sd-v2.1-base-4view. To produce diverse 3D representations, we adapt our algorithm to generate NeRF outputs and follow the geometry and mesh refinement stages as outlined in ProlificDreamerWang et al. (2024). Additionally, we employ the Gaussian Splatting generation pipeline from GaussianDreamerYi et al. (2024), resulting in 3D NeRF, mesh, and Gaussian Splatting outputs. To align the generated 3D content with human preferences, we integrate two human RLHF models: ImageRewardXu et al. (2023) and PickScoreKirstain et al. (2023).

For PickScore, we use a scale of 100 for mesh generation and 10,000 for Gaussian Splatting generation. Empirically, we found that a high PickScore scale enhances visual richness but may lead to color oversatura-

tion. The optimal PickScore scale ranges between 50–200 for mesh generation and 5,000–10,000 for Gaussian Splatting generation.

**Score-based Divergence Implementation**   For the score-based loss in Equation 15, we use the squared $l_2$ norm as the distance function $d$, where $d(s_p, s_q) = \|s_p - s_q\|_2^2$. The sampling distribution $\pi_t$ is simply the marginal distribution $p(x_t|x_0)$ from the forward diffusion process applied to the current rendered image $x_0 = g(\theta, c)$. This ensures that the score matching occurs in the data-relevant regions of the diffusion trajectory. It is important to note that our method **does not require training a separate score approximation network**. Instead, it comprises the frozen weights of the original score network $s_\phi$ with trainable LoRA layers injected into its attention blocks, which is exactly the same as ProlificDreamer Wang et al. (2023c). The LoRA parameters $\varphi$ are trained concurrently with the 3D representation parameters $\theta$ at each optimization step. The objective function for training $\varphi$ is the standard denoising score matching (DSM) loss. At each step, using the rendered image $x_0 = g(\theta, c)$, we compute the noisy image $x_t = \alpha_t x_0 + \sigma_t \epsilon$. The DSM loss then minimizes the difference between the network's score prediction and the true score of the noisy data distribution, which is known to be $-\epsilon/\sigma_t$. This training step corresponds to Line 33 in our pseudo-code (Algorithm 1) and ensures that $s_\varphi$ provides an accurate, up-to-date estimate of the score of the generator's current output distribution, which is essential for the SIM loss calculation.

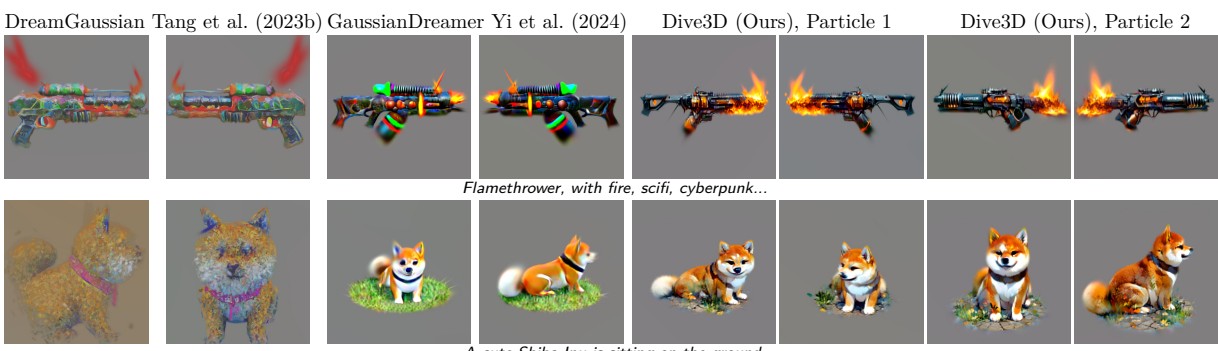

Figure 6: **Comparison with baselines on Gaussian Splatting.** Dive3D can generate diverse and human-preferred 3D Gaussian splattings.

# D   Ablation Study

## D.1   SIM vs. Reward Guidance

To disentangle the contributions of our score-based divergence (SIM) and reward guidance, we conduct an ablation study. As shown in Table 4, the results demonstrate that switching from a KL-based objective to SIM is the primary driver for increased diversity (CLIP-D). Subsequently, adding the reward term is key to achieving state-of-the-art quality and text-alignment (ImageReward score), clarifying the distinct role of each component.

Table 4: Ablation study on the components of Dive3D.

| Method | Loss Formulation | Diversity (CLIP-D) ↑ | Quality (ImageReward) ↑ |
| --- | --- | --- | --- |
| Baseline (ProlificDreamer) | $\mathcal{L}_{KL}$ | 0.3908 | 0.802 |
| Dive3D (No Reward) | $\mathcal{L}_{SIM}$ | **0.4412** | 0.839 |
| **Dive3D (Full)** | $\mathcal{L}_{SIM}$ + Reward | **0.4483** | **0.977** |

## D.2   Impact of Divergence Loss Combinations

In this section, we investigate how different combinations of text-to-3D divergence losses affect generation performance. Our framework involves three fundamental divergence losses: the CDP loss, the UDP loss,

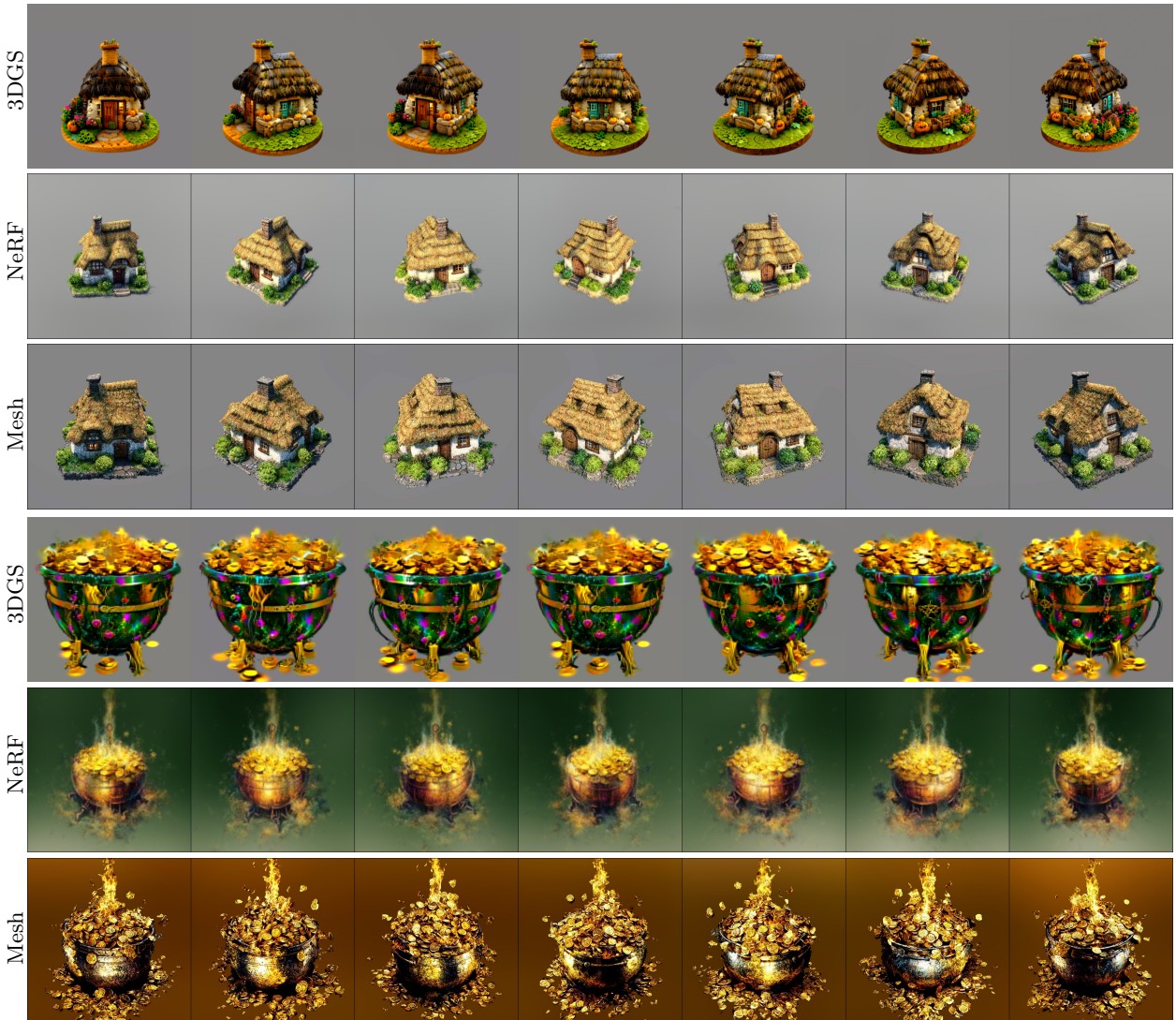

Figure 7: **Dive3D can generate 3D objects across different 3D representations.** We show the results on Gaussian splitting, NeRF and Mesh. The prompts are *A 3D model of an adorable cottage with a thatched roof* and *A cauldron full of gold coins.*

and the ER loss.

$$\mathcal{L}_{\text{Dive3D}} = \alpha_1 \mathcal{L}_{\text{CDP}} - \alpha_2 \mathcal{L}_{\text{UDP}} + \alpha_3 \mathcal{L}_{\text{ER}} \tag{33}$$

We systematically explore the effects of different loss weights by analyzing all possible paired combinations.

**CDP and UDP** As shown in Fig. 9(a), assigning a large weight to the CDP loss preserves generation diversity but results in blurred outputs, while a large weight on the UDP loss introduces significant artifacts, preventing the 3D generation from capturing basic image features. Empirically, the optimal performance is achieved when the weights for CDP and UDP are balanced, i.e., when the ratio $\alpha_1/\alpha_2$ is approximately one. This finding aligns with the observations for tuning the CFG loss(Wang et al., 2024), which typically attains the best performance when the key component, a balanced linear combination of CDP and UDP losses, is strongly emphasized.

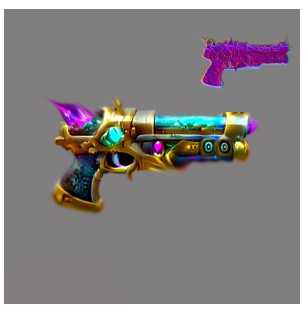 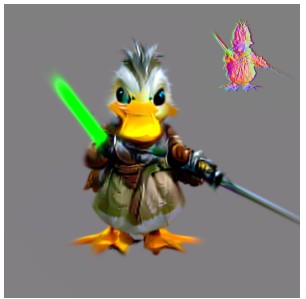 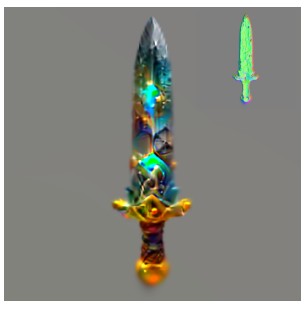 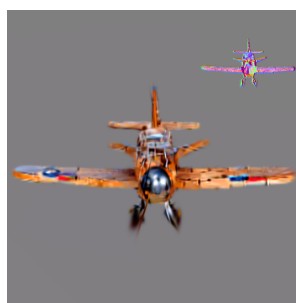

*Magic gun, game asset, mystery.*   *Jedi duck holding a lightsaber.*   *Magic dagger, mystery, ancient.*   *An airplane made out of wood.*

Figure 8: **Dive3D 3D Gaussian Splattings.** Dive3D can generate anime-style and game-style 3D Gaussian Splattings.

**CDP and ER**   We employ two human-preference reward models, ImageReward (Xu et al., 2023) and PickScore (Kirstain et al., 2023), in our experiments. As shown in Fig. 9(b), in both cases, higher ER scores lead to improved 3D generation by enhancing text alignment and revealing finer textual details. Different reward models can guide generation in distinct ways; for instance, in our test case, ImageReward and PickScore exhibit different preferences for color combinations that evoke a sense of mystery in the generated game assets.

**UDP and ER**   We further explore the combination of UDP and ER losses—a configuration not previously investigated in text-to-3D generation. As shown in Fig. 9(c), this novel combination also produces reasonable 3D objects, demonstrating that text-to-3D generation is possible without relying on a text-to-image model. However, we observe that its performance is highly sensitive to the initialization of the 3D representation; poor or biased initialization can trap the algorithm in local minima in the absence of text-to-image guidance. Consequently, although the ER+UDP combination is promising, integrating all three basic losses yields the best overall results.

### D.3   Hyperparameter Design and Selection

The weights for the core divergence terms in Equation 33 ($\alpha_1$, $\alpha_2$, $\alpha_3$) were determined empirically on a validation subset of 10 prompts. Our goal was to balance text-alignment, realism, and diversity. The systematic exploration shown in Figure 9 of the main paper provides a detailed visualization of this process. For the final model, we found that setting the ratio $\alpha_1/\alpha_2$ close to 1 (consistent with the $(1 + \gamma)$ and $-\gamma$ formulation) and a reward weight $\lambda$ (corresponding to $\alpha_3$) that enhances detail without causing oversaturation yielded the best results. The final values used are consistent with the principles illustrated in that analysis.

## E   More Results

We provide more results in Figure 6, 7 and 8.

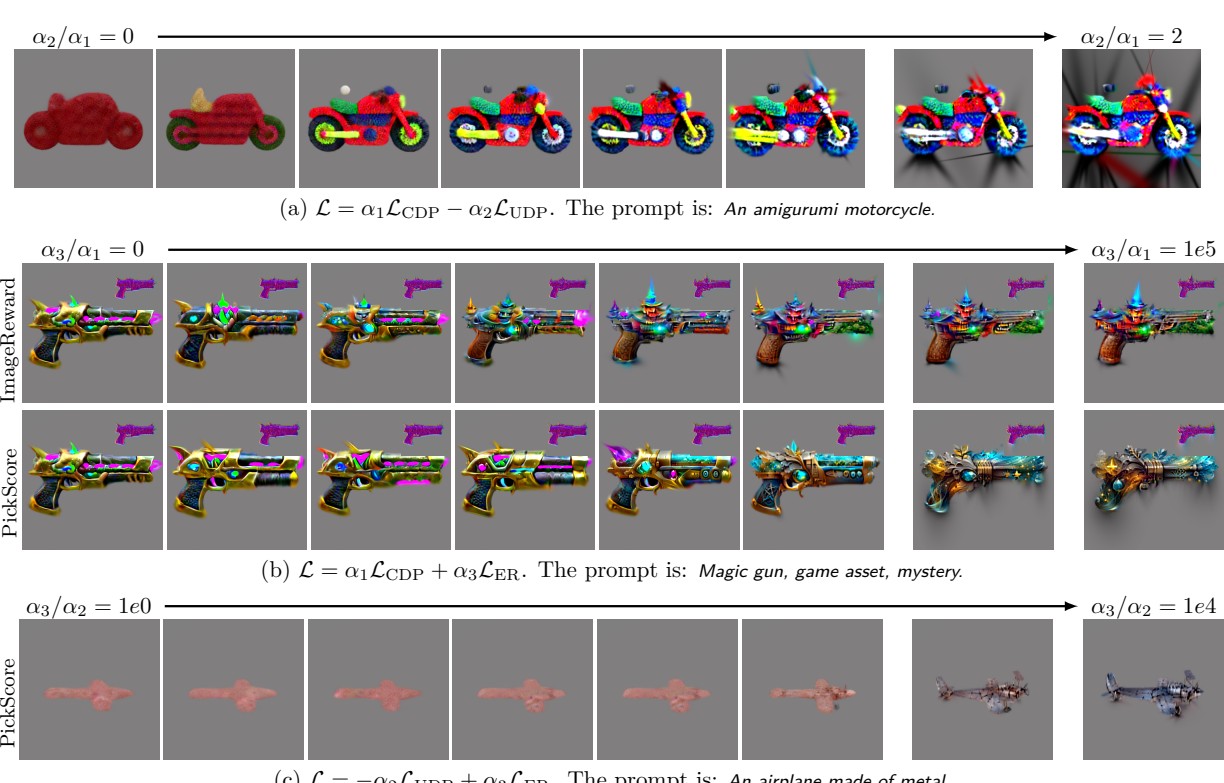

(a) $\mathcal{L} = \alpha_1 \mathcal{L}_{\mathrm{CDP}} - \alpha_2 \mathcal{L}_{\mathrm{UDP}}$. The prompt is: *An amigurumi motorcycle.*

(b) $\mathcal{L} = \alpha_1 \mathcal{L}_{\mathrm{CDP}} + \alpha_3 \mathcal{L}_{\mathrm{ER}}$. The prompt is: *Magic gun, game asset, mystery.*

(c) $\mathcal{L} = -\alpha_2 \mathcal{L}_{\mathrm{UDP}} + \alpha_3 \mathcal{L}_{\mathrm{ER}}$. The prompt is: *An airplane made of metal.*

Figure 9: **Exploration of the design space of the unified divergence loss.** We investigate their influence of different text-to-3D divergence losses on generation performance by adjusting their weights. 3D Gaussian splitting are used as representations in these experiments for computational efficiency.

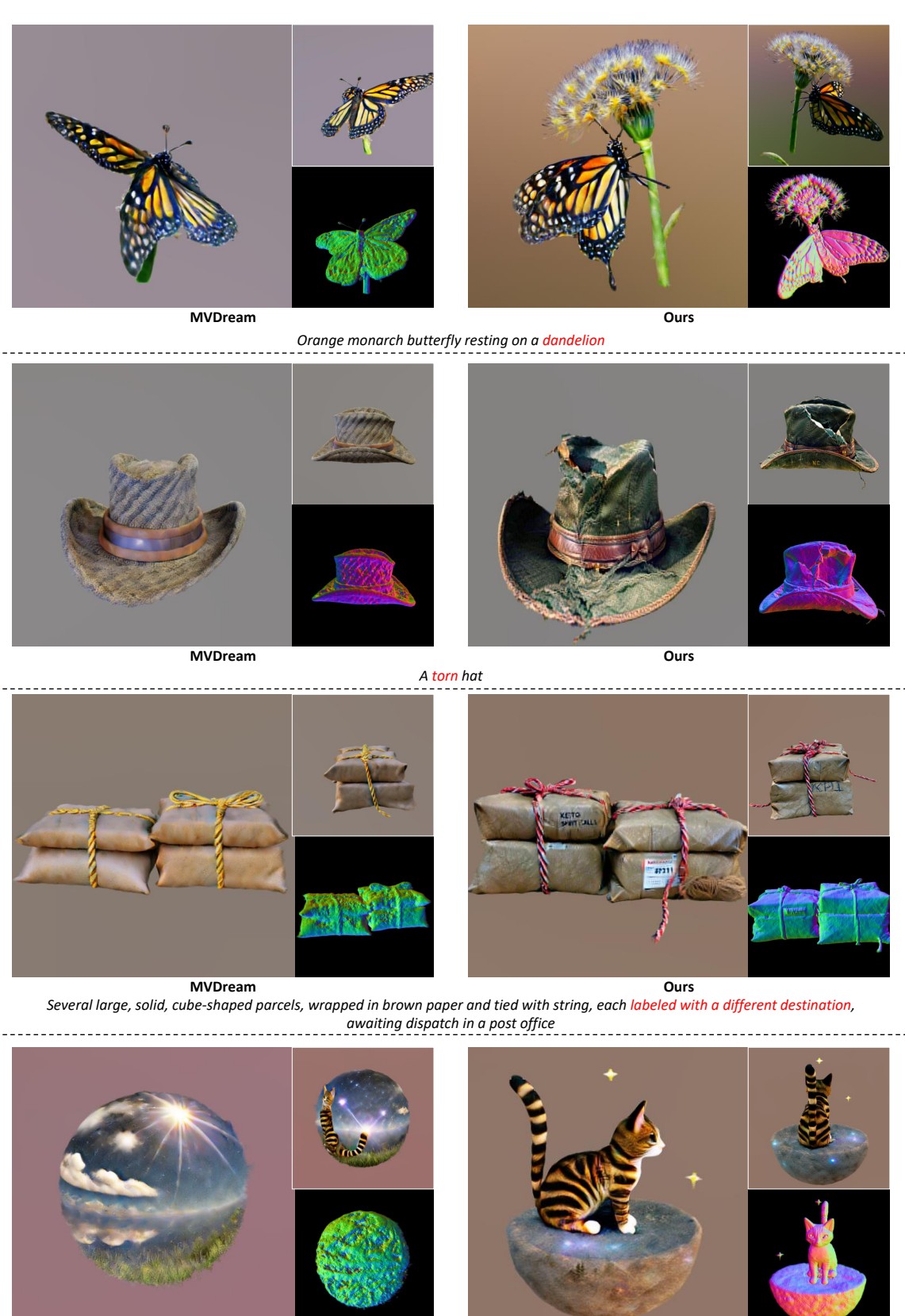

Figure 10: More comparison results of Dive3D, showing improved quality over prior methods. All results are based on the MVDream pipeline Shi et al. (2023) using Multi-view Diffusion Shi et al. (2023).

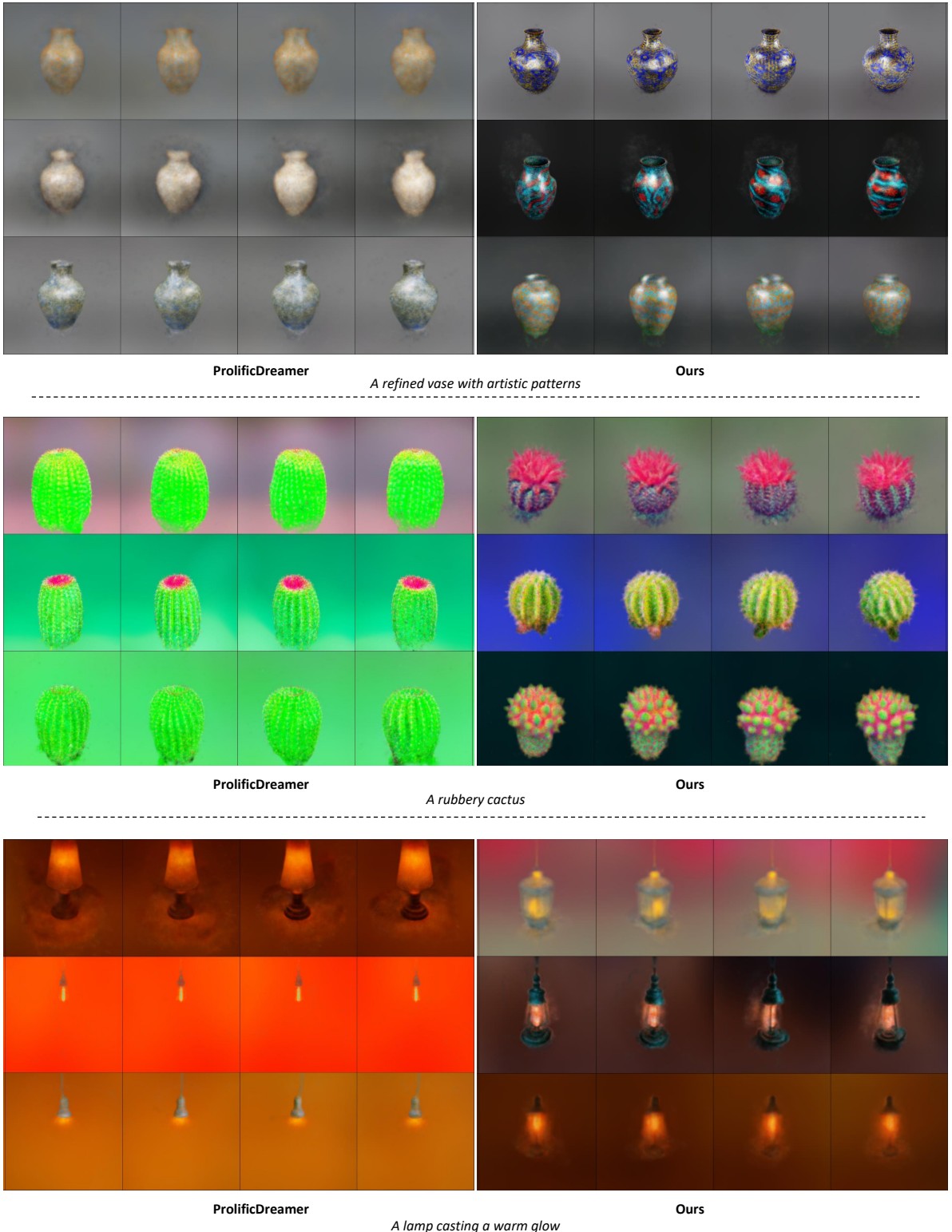

Figure 11: More comparison results of Dive3D, showing improved diversity over prior methods. All results are based on Stage 1 of the ProlificDreamer pipeline Wang et al. (2023c) using Stable Diffusion Rombach et al. (2022), without further refinement.

