# OpenReview forum: "Dive3D: Diverse Distillation-based Text-to-3D Generation via Score Implicit Matching"
_TMLR — Accepted by TMLR_

### Review · Reviewer_oNEN · 2025-10-04

**Summary Of Contributions:**

The paper reformulates both Score Distillation Sampling (SDS) and reward-based text-to-3D methods as linear combinations of three KL divergence terms, revealing their shared mathematical foundation and common limitations in mode-seeking. To address this, the authors replace KL objectives with Score Implicit Matching (SIM) loss, which matches score functions rather than probability densities, thereby claiming to avoid mode collapse. The method achieves state-of-the-art performance on the GPTEval3D benchmark (top scores across all six metrics). It demonstrates qualitative improvements in both diversity and quality against nine baselines across multiple 3D representations.

Strengths:
- The KL divergence unification (Equations 10-14) provides valuable conceptual clarity, showing that different methods share underlying objectives.
- Comprehensive evaluation showing consistent improvements in quality, text alignment, and diversity.
- Mode collapse in text-to-3D is a fair limitation that impacts practical applications.
- Works across multiple 3D representations (NeRF, mesh, Gaussian Splatting) and diffusion backbones.

Weaknesses:
- Claims SIM "avoids mode collapse" without formal proof. Section 4.3 provides intuition rather than mathematical justification.
- Only 16 prompts for quantitative diversity assessment. This paper lacks geometric diversity metrics.
- No runtime or memory comparisons with baselines, despite this being critical for practical adoption.
- Distance function d, sampling distribution \pi_t, and score approximation network training are unspecified.

**Audience:**

Yes

**Audience Explanation:**

The paper addresses mode collapse in text-to-3D generation, which is a timely problem for the ML and CV communities. The unified framework, revealing that SDS and reward methods share KL-based foundations, provides valuable conceptual insight. The empirical finding that score-based divergence improves both diversity and quality is relevant to researchers in diffusion models, 3D generation, and generative modeling.

**Broader Impact Concerns:**

No concerns.

**Claims And Evidence:**

No

**Claims Explanation:**

The central claim that SIM "mitigates mode collapse" lacks formal proof. Section 4.3 provides intuition rather than mathematical justification. The diversity claim is evaluated on only 16 prompts with two metrics (CLIP-D, FID), which is insufficient for claims of "significantly more diverse" outputs. Critical information is missing: no computational cost comparison and unspecified implementation details (distance function d, sampling distribution \pi_t). In the meantime, GPTEval3D results and qualitative comparisons convincingly demonstrate quality improvements.

**Requested Changes:**

- Either provide formal proofs that SIM avoids mode collapse, or reframe all claims as empirical observations rather than theoretical guarantees. Throughout the paper (abstract, introduction, Section 4.3, conclusion), replace phrases like "effectively mitigates mode collapse" and "avoids mode-seeking tendencies" with "empirically demonstrates reduced mode collapse" and "shows less mode-seeking behavior in experiments."
- Increase quantitative diversity assessment to at least 25-50 prompts across diverse categories. Add geometric diversity metrics specific to 3D (e.g., Chamfer distance between meshes for the same prompt). Include statistical significance testing (t-tests, p-values) for CLIP-D and FID improvements. Report complete user study protocol.
- Add runtime comparisons with baselines, memory overhead measurements, and a computational cost table showing: Method | Time (hours) | GPU Memory (GB) | alongside quality/diversity scores.
- Specify the distance function d used in Equation 15, the sampling distribution \pi_t, and the score approximation network architecture and training procedure.

---

### Review · Reviewer_xLMJ · 2025-10-06

**Summary Of Contributions:**

This paper presents Dive3D, a novel text-to-3D generation framework designed to address the limited diversity in existing diffusion distillation and reward-based text-to-3D methods. The key innovation lies in replacing the conventional Kullback–Leibler (KL) divergence objectives (used in Score Distillation Sampling, SDS, and related methods) with a Score Implicit Matching (SIM) loss. This score-based divergence encourages exploration of multiple high-probability regions, alleviating the mode-seeking tendencies inherent to KL minimization. The authors further unify diffusion-based and reward-based paradigms under a single divergence-based framework, enabling principled integration of diffusion priors, human preference signals, and diversity-promoting objectives. Experimental results demonstrate that Dive3D achieves state-of-the-art performance.

Strengths:
- This paper is well-motivated. The limited diversity and the asymmetric nature KL divergence are effectively recognized.
- The proposed theoretical validation further supports the effectiveness and rationale of the proposed method.
- This paper unifies diffusion distillation and proposes a reward-based optimization into a single divergence-based formulation, which can further simplify the training paradigm.
- The experimental evaluation is quite extensive, and various visualizations under different scenarios are presented, which validates the effectiveness of the proposed method.

Weaknesses:
- Lack of ablation analyses of the reward and Dive3D objectives. There are multiple losses proposed to achieve the final performance, but they are not theoretically nor empirically understood. At the same time, it is unclear how the hyperparameters are chosen. Thus, the reproducibility of the paper would be difficult.
- Lack of computational efficiency justification. Since SIM may require additional sampling or gradient estimation steps, it would be useful to report training time and computational overhead relative to SDS-based methods.
- The paper claimed the diversity of the generation is one vital problem; however, it does not specify how to quantify whether the proposed method can indeed enhance generation diversity. Thus, it would be beneficial to incorporate various diversity metrics, such as feature variance, LPIPS across samples, or FID diversity, to justify the contribution.

**Audience:**

Yes

**Audience Explanation:**

The research topic is attention-drawing, and the performance is quite promising.

**Broader Impact Concerns:**

No ethical issues.

**Claims And Evidence:**

Yes

**Claims Explanation:**

Most of the claims are well-supported.

**Requested Changes:**

Please see the weaknesses part.

---

### Review · Reviewer_DvKe · 2025-10-07

**Summary Of Contributions:**

This paper proposes to tackle the problem of diversity in text-to-3D generation. They propose to use Score Implicit Matching instead of KL divergence to avoid model collapse. This allows for a better diversity since the loss isn't stuck in a high-density area of the KL divergence.
They first propose a unified KL divergence framework, which is a linear combination of three losses: one for prompt fidelity, one for diversity, and one for user reward. Then they change to score-based divergence instead of KL divergence.
To test their new method called Dive3D, they did a benchmark on GPTEval3D compared to 10 other methods. Then they propose a analysis of generation diversity.

**Audience:**

Yes

**Audience Explanation:**

Dealing with diversity in text-to-3D generation is a problem that has been raised and studied in many papers in the field. The proposed method improves diversity, making this solution a good finding for the community.

**Claims And Evidence:**

Yes

**Claims Explanation:**

The paper clearly explains the fundamental problem in text-to-3D generation. Most of the methods struggle to have high diversity in their generated images. This problem is clearly explained in the paper, and the authors proposed to use score-based divergence instead of Kl divergence to be able to have more flexibility in the generation. The claim of having better variability is proven in the experimental part where Dive3D outperforms 10 competitors on 5 different scores. In addition, the authors propose an experiment that focuses on diversity analysis. This study shows that Dive3D generated more diverse images.

**Requested Changes:**

- Dive3D proposes two improvements compared to the literature: expression of the loss in three terms: prompt quality, diversity, and reward, and use of the Score-based divergence. It could be nice to have an ablation study to see what helps the best for diversity and/or other scores of the benchmark.

- One limit raised by the authors is the time computation of their method. For transparency, adding a comparison of the runtime would be a good addition. It can allow user to know if Dive3D is scalable for their application.

---

### Decision · Action_Editor_7fAs · 2025-12-17

**Recommendation:** Accept as is

**Audience:**

Yes

**Audience Explanation:**

Several reviewers explicitly state that the findings would interest and benefit the TMLR community, especially those working on generative modeling beyond pure 2D synthesis.

**Claims And Evidence:**

Yes

**Claims Explanation:**

Multiple reviewers agree that the paper provides strong empirical evidence that Dive3D improves both diversity and quality in text-to-3D generation.  Initial concerns about unclear contributions of SIM vs. reward guidance, lack of runtime analysis, and missing diversity quantification were fully addressed.

One reviewer initially questioned the lack of formal proof that SIM mitigates mode collapse. In response, the authors added a formal theoretical analysis, demonstrating why KL divergence is mode-seeking while SIM is not, thereby strengthening the paper’s central claim.